# KLF5 loss sensitizes cells to ATR inhibition and is synthetic lethal with *ARID1A* deficiency

Samah W. Awwad [1,2] ✉, Colm Doyle[1], Josie Coulthard[1], Aldo S. Bader [1,2], Nadia Gueorguieva[1,2], Simon Lam [1,2], Vipul Gupta[2], Rimma Belotserkovskaya[1,2], Tuan-Anh Tran[1], Shankar Balasubramanian[1,3] & Stephen P. Jackson [1,2,4] ✉

ATR plays key roles in cellular responses to DNA damage and replication stress, a pervasive feature of cancer cells. ATR inhibitors (ATRi) are in clinical development for treating various cancers, including those with high replication stress, such as is elicited by ARID1A deficiency, but the cellular mechanisms that determine ATRi efficacy in such backgrounds are unclear. Here, we have conducted unbiased genome-scale CRISPR screens in *ARID1A*-deficient and proficient cells treated with ATRi. We found that loss of transcription factor KLF5 has severe negative impact on fitness of *ARID1A*-deficient cells while hypersensitising *ARID1A*-proficient cells to ATRi. KLF5 loss induced replication stress, DNA damage, increased DNA-RNA hybrid formation, and genomic instability upon ATR inhibition. Mechanistically, we show that KLF5 protects cells from replication stress, at least in part through regulating BRD4 recruitment to chromatin. Overall, our work identifies KLF5 as a potential target for eradicating *ARID1A*-deficient cancers.

Accurate genome duplication during S phase is imperative for faithful cell cycle progression and genome integrity. Nevertheless, the DNA replication machinery faces various perturbations, arising through the actions of both exogenous and endogenously arising agents, that may cause slowing or stalling of DNA replication forks and associated "replication stress". Once triggered, replication stress leads to the accumulation of single-stranded DNA (ssDNA), which is recognised and bound by replication protein A (RPA), thus creating a signal for activating the replication stress response (RSR)[1,2]. In eukaryotic cells, the apical regulators of the RSR are ATR (Ataxia Telangiectasia and Rad3-related) and its major downstream target, CHK1 (checkpoint kinase 1). ATR and CHK1 are protein kinases that play key roles in cellular responses to DNA damage, including double-stranded breaks (DSBs), DNA replication stress, and accumulation of ssDNA or ssDNA gaps[3–7].

ATR is normally essential for cell proliferation and responds to replication stress in various ways to preserve genome integrity, including by stabilising and promoting the restart of stalled DNA replication forks, controlling DNA replication-origin firing, and restricting cells with defective DNA replication from premature entry into mitosis. Indeed, in the absence of ATR function, stalled replication forks are converted into cytotoxic DSBs through replication-fork collapse—events that threaten genomic stability and cell viability[4]. ATR and CHK1 also play pivotal roles in unperturbed S phase by regulating dormant replication-origin firing and controlling the S/G2 as well as G2/M cell cycle transitions[4,8].

Notably, replication stress has been identified as a hallmark of human cancer cells[1,9]. While this may in-part be due to faster proliferation rates and deoxyribonucleotide imbalances in S phase, amplification of oncogenes such as *CCNE1* and *MYC* induces replication stress by shortening G1 and promoting uncoordinated firing of intragenic replication origins that would otherwise be repressed by near-completed transcription[10]. This also increases conflicts between the replisome and the transcription machinery, associated with formation of DNA-RNA hybrids (R-loops)[11], leading to genome instability and threatening cell survival through replication-fork collapse[12,13]. The activity and functions of ATR are therefore normally crucial for cell viability, mainly in highly proliferative cells including cancer cells, due to generation of replication stress and replication-driven DNA damage.

[1]Cancer Research UK Cambridge Institute, University of Cambridge, Cambridge, UK. [2]The Gurdon Institute, University of Cambridge, Cambridge, UK. [3]Yusuf Hamied Department of Chemistry, University of Cambridge, Cambridge, UK. [4]Department of Biochemistry, University of Cambridge, Cambridge, UK. ✉e-mail: samah.diab@cruk.cam.ac.uk; steve.jackson@cruk.cam.ac.uk

Importantly, ATR inhibitors (ATRi) have been developed and are currently in advanced clinical trials for treating various cancers[14], including those with defects in DNA damage response (DDR) factors such as ATM[15–17] and ERCC1[18]. Moreover, through employing high-throughput CRISPR-Cas9 and RNAi screening approaches, we and others have identified various genetic and functional biomarkers for ATRi efficacy[19–24].

Notably, loss-of-function mutations in ARID1A (AT-rich interactive domain containing protein 1A)—the subunit of the canonical BAF (BRG1/BRM-associated factor) chromatin remodelling complex that is most often perturbed in human cancers[25–27]—have been proposed as candidates for affecting ATRi sensitivity[28]. At present, however, it remains unclear which functions and pathways dominate ATRi efficacy in clinical settings, and whether and how ATRi resistance might arise in cells lacking ARID1A. Elucidating common and genetic-background specific mechanisms of ATRi efficacy and resistance could thus assist patient stratification and might also suggest ways to circumvent or even exploit ATRi resistance.

Herein, we employ genome-scale CRISPR-Cas9 screens in ARID1A-deficient and proficient cells combined with ATRi. We find that loss of the transcription factor, KLF5 (Krüppel-like transcription factor 5) causes hypersensitivity to ATRi in ARID1A-proficient, but not in ARID1A-deficient cells; a distinction explained by our observation that KLF5 is needed for proliferation of ARID1A-deficient cells. We provide evidence that KLF5 loss causes transcription-dependent replication stress accompanied by increased DNA-RNA hybrid formation and genomic instability upon ATR inhibition. Functionally, we show that KLF5 protects from replication stress, at least in part through regulating BRD4 chromatin recruitment. Collectively, our data reveal a hitherto unrecognised role of KLF5 in promoting genome stability and suggest KLF5 as a therapeutic target for ARID1A-deficient tumours.

## Results

### Factors affecting ATRi sensitivity in ARID1A-proficient or -deficient cells

To systemically explore mechanisms of ATRi sensitivity and/or resistance and identify potential genetic dependencies for ARID1A-deficient cells, we performed unbiased genome-scale CRISPR genetic screens with an isogenic pair of ARID1A wild-type (WT) and ARID1A-knockout (KO) human U2-OS cells that stably express active Streptococcus pyogenes Cas9 (spCas9) nuclease (Supplementary Fig. 1a, b). Following transduction with a lentiviral single-guide RNA (sgRNA) library, each cell population was divided into two and propagated for 15 days under control conditions (DMSO) or in the presence of AZD6738 (aka ceralasertib, a potent ATRi in clinical trials)[29,30] at an IC$_{50}$ dose (inhibitory concentration causing 50% reduction in cell number) pre-calculated for each genetic background. DNA was extracted from the surviving cell pools, as well as the pre-treated cells, and the regions encoding sgRNAs were PCR amplified and subjected to next-generation DNA sequencing (Fig. 1a).

Following successful quality control checks of screen outputs (Supplementary Fig. 1c), ensuing bioinformatics analyses using DrugZ revealed (by comparing raw sgRNA counts in ATRi-treated cells with those from DMSO-treated cells) various previously described factors and pathways affecting ATRi efficacy in WT cells. These included ATM[17,31,32], RNASEH2A/C, and the DNA polymerase ε accessory subunits POLE3 and POLE4[20,24] (Fig. 1b and Supplementary Data). Furthermore, our analyses identified genes whose deficiencies are known to promote ATRi-resistance in WT cells, such as CDC25A/B[23]. Collectively, these screen hits served as positive controls, indicating the reliability of our screening approach.

Moreover, we identified intriguing differences in gene hit profiles between the ARID1A WT and ARID1A KO backgrounds (Fig. 1b, c, respectively; with differential analysis displayed in Fig. 1d). For instance, our screens revealed that RNASEH2A/C, KLF5 and SMARCA4

behaved as strong dropouts (genes that when depleted, are associated with increased drug sensitivity) in ARID1A WT cells, while they were not or only mildly depleted in ARID1A KO cells (Fig. 1d). These data therefore implied that loss of RNASEH2A, RNASEH2C, KLF5, or SMARCA4 selectively hypersensitize ARID1A WT, but not ARID1A KO cells, to ATRi. Conversely, we found some ATRi dropouts in the ARID1A KO background but not in ARID1A WT cells, such as FOXD4L4, LARP4 and SPEF2 (Fig. 1d). In addition, some hits showed similar behaviour in WT and ARID1A-deficient cells, such as the oxidative stress sensor, KEAP1, which overall behaved as the strongest gene dropout in our screens. Collectively, these data reflected successful systematic identification of genes and/or pathways affecting cellular responses towards ATR inhibition by AZD6738 and highlighted how their effects can vary depending on cellular ARID1A status.

Furthermore, analysis of our data derived from screens carried out under control (DMSO-treated) conditions, revealed several genetic dependencies selective for ARID1A KO cells. For example, ARID1B loss was identified as impairing survival of ARID1A KO but not WT cells (Fig. 1e). This genetic interaction is consistent with previous findings showing that ARID1B is crucial for ARID1A KO but not WT cells, presumably reflecting the overlapping and partly redundant roles of ARID1A and ARID1B in the essential canonical BAF (cBAF) complex[33–36]. Notably, our analyses also highlighted KLF5—which encodes Krüppel-like transcription factor 5—as a gene whose inactivation decreased proliferation of ARID1A KO but not ARID1A WT cells (Fig. 1e). To validate this, we depleted KLF5 in WT and ARID1A KO U2-OS cells using two different siRNAs and assessed cell proliferation by colony formation assays. As shown in Fig. 1f, while KLF5 depletion in WT cells did not noticeably affect their colony-forming ability, it markedly impaired colony formation of ARID1A KO cells.

To further corroborate these findings, we generated ARID1A KO RPE-1 cells and assessed the impact of KLF5 depletion in these and in WT RPE-1 cells. In agreement with the results obtained using U2-OS cell line (Fig. 1f), we observed that siRNA-mediated depletion of KLF5 impaired the survival of three different ARID1A KO clones but not the survival of WT RPE-1 cells (Supplementary Fig. 1d). Additionally, we took advantage of the commercially available KLF5 inhibitor (ML264) and assessed its effect on the proliferation of WT and ARID1A KO U2-OS and MCF7 cells. In both cell backgrounds, we observed that KLF5 inhibition caused a greater reduction in growth of ARID1A mutant cells than the ARID1A WT controls (Supplementary Fig. 1e, f). Furthermore, we utilised the triple negative breast cancer cell line, CAL-51, which does not express ARID1A, complemented these cells with vectors expressing GFP or GFP-ARID1A WT (Supplementary Fig. 1g) and assessed the effect of KLF5 inhibition on their growth. Thus, we observed that complementation with GFP-ARID1A WT rescued the growth defect observed upon KLF5 inhibition in CAL-51 cells (Supplementary Fig. 1h). Collectively, these observations both further validate the results of our screen and demonstrate that the observed effects are not cell line specific.

The aforementioned experiments reveal a previously undescribed synthetic lethal relationship between KLF5 and ARID1A. To explore this further, we used DepMap data[37] to examine the effect of CRISPR-mediated KLF5 gene knockout in ARID1A-deficient cancer cell lines. As shown in Supplementary Fig. 1i, targeting KLF5 in ARID1A-deficient cancer cells has a negative CRISPR score, which indicates ARID1A mutant cells are sensitive to KLF5 depletion. As a positive control, we examined the effect of targeting ARID1B, a known synthetic lethal partner of ARID1A, which showed a negative CRISPR score. Notably, KLF5 is a transcription factor that belongs to the KLF family, which in human cells consists of 17 members, thus, we assessed the effect of other transcription factors: KLF4 as another member of the KLF family, and E2F7 as a more broadly used transcription factor. This indicated that targeting either KLF4 or E2F7 does not appear to affect ARID1A mutant cell lines (Supplementary Fig. 1i). These analyses further strengthen our observations

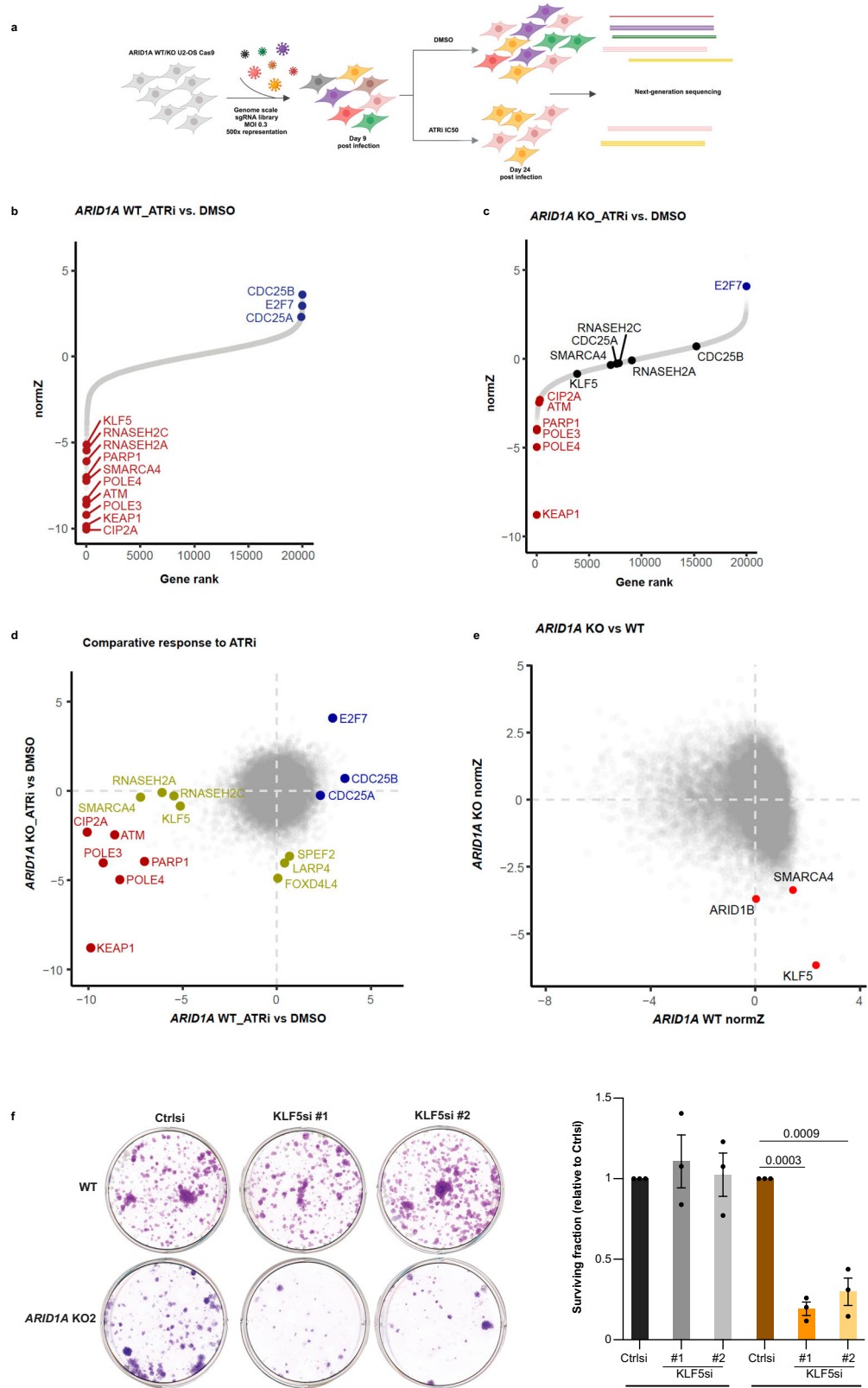

and conclusions that KLF5 is a synthetic lethal target for *ARID1A*-deficient cells and implies the specificity of KLF5 in such settings.

## KLF5 loss hypersensitizes cells to replication stress

For ensuing mechanistic studies, we focused on *KLF5*. Firstly, we depleted KLF5 by using siRNA in U2-OS cells (Supplementary Fig. 2a)

and subjected cells to clonogenic survival assays in the presence or absence of ATRi. As shown in Supplementary Fig. 2b, cells treated with siRNA targeting KLF5 were more sensitive to ATRi treatment compared to control cells. Secondly, we generated (using CRISPR-Cas9 mediated gene disruption) *KLF5* KO human U2-OS, RPE-1, and HAP-1 cell lines (Fig. 2a and Supplementary Fig. 2c–e), and observed that in all cell lines

**Fig. 1 | CRISPR-Cas9 screens for factors affecting ATRi sensitivity in WT and *ARID1A* KO U2-OS cells. a** Schematic of genome-scale CRISPR-Cas9 screen layout. WT and *ARID1A* KO (clone 2 and 24) U2-OS cells stably expressing Cas9 were infected with the genome-wide Brunello library at a multiplicity of infection (MOI) 0.3 at 500× representation. 48 h post infection, cells were selected with puromycin for a further 6 days. Cells were then treated with ATRi (AZD6738) or DMSO. After 14 days, cells were harvested, the abundance of each sgRNA within the different samples was quantified by next-generation sequencing, and DrugZ software was used to compare read-counts between the different conditions. Created in BioRender. Diab, S. (2025) https://BioRender.com/y46l947. **b, c** Rank plots of genes affecting ATRi sensitivity based on DrugZ analysis of results from CRISPR-Cas9 screening in *ARID1A* WT (**b**), and *ARID1A* KO (**c**) U2-OS cells; normZ scores show differential effects of gene knockouts with the ATRi compared to DMSO. Selected genes with normZ score of less than (−2) were designated as hypersensitizing hits and are shown in red. Selected genes with normZ score greater than zero were designated resistance hits and are shown in blue. **d** Biplot for combinational comparison of ATRi modifier genes between WT (*x* axis) and ARID1A KO (*y* axis) U2-OS cells; *z*-scores from A and B were used. Differential hits are shown in gold. **e** Biplot of *ARID1A* co-essential genes based on DrugZ analysis. Read-counts for DMSO-treated *ARID1A* KO and WT cells were compared to those of the starting population by using DrugZ separately. **f** Representative images and quantification of the colony-forming ability of WT and *ARID1A* KO2 U2-OS cells upon KLF5 siRNA depletion (biological *n* = 5). Graphs are depicted with means ± SEM. Statistical analyses were performed using one-way ANOVA with multiple comparisons. Source data are provided as a Source Data file.

tested, loss of KLF5 hypersensitized cells to ATRi (Fig. 2b–d). To establish whether the observed phenotype was indeed due to losing KLF5, we complemented *KLF5* KO U2-OS cells with vectors expressing GFP only or GFP-KLF5 WT and found that restoring KLF5 expression rescued ATRi sensitivity (Fig. 2e). As a complementary approach, we treated U2-OS cells with combinations of KLF5 inhibitor (ML264) and ATRi (AZD6738) and observed that the compounds had additive effects on cell survival (Supplementary Fig. 2f).

ATR inhibition leads to replication stress and activates the replication stress response (RSR). To further explore whether the effect of KLF5 depletion was general to replication stress or specific to the RSR, we performed clonogenic cell survival assays in the presence of an inhibitor of CHK1 (Fig. 2f and Supplementary Fig. 2g), which plays key roles in the RSR, or in the presence of replication stress-inducing agents such as hydroxyurea (Fig. 2g) and camptothecin (Fig. 2h). KLF5 loss hypersensitized cells to each of these agents, but not to the DNA-DSB inducing agent, etoposide (VP16; Supplementary Fig. 2h). These findings implied that loss of KLF5 selectively hypersensitizes cells to replication stress-inducing agents.

## Loss of KLF5 induces DNA damage and genomic instability upon ATRi

To investigate the mechanisms by which KLF5 loss sensitizes cells to ATRi, we examined cell-cycle profiles of WT and *KLF5* KO cells upon ATRi treatment. To this end, we labelled cells with the nucleotide analogue EdU and stained them with DAPI for DNA content and used flow cytometry to assess the proportions of cells in each cell cycle phase. We observed that *KLF5* KO cells have higher percentage of cells in S phase (EdU positive) when compared to WT cells in untreated conditions, in both U2-OS and RPE-1 cells (Fig. 3a, b and Supplementary Fig. 3a, b). This suggests that *KLF5* KO cells experience higher replication stress, which is sensed by the intra-S phase checkpoint, and thus progress slower through S phase. However, upon treatment with ATRi, both WT and *KLF5* KO U2-OS cells accumulated in G1 phase and became reduced in the proportion of cells in S phase (Fig. 3a). These observations were in line with the role of ATR in regulating the intrinsic S-G2 checkpoint to ensure complete DNA replication before cells enter G2 phase[8], and ATR's role in supporting DNA replication in early S phase[38], which is why cells treated with ATRi accumulate in G1. Moreover, we observed similar effects on cell cycle distribution upon ATRi treatment in *KLF5* KO in RPE-1 cells, whereas RPE-1 WT cells showed a slight increase of cells in G1 and no significant difference in the proportion of cells in S phase upon ATRi treatment (Supplementary Fig. 3a). Such difference between parental U2-OS and RPE-1 might be because of different genetic backgrounds between cell lines.

In addition, we observed that loss of KLF5 led to accumulation of more DNA damage as measured by γH2AX (Ser-139 phosphorylated histone H2AX; a widely used marker for DNA DSBs) that occurred mainly in S phase, upon ATRi or CHK1i treatment (Fig. 3c–f). Expectedly, these γH2AX-positive cells displayed lower EdU intensity in comparison to the S phase γH2AX-negative cells, reflecting the lesions caused by ATRi treatment impairing productive replication (Supplementary Fig. 3c).

The observed increase in the percentage of γH2AX-positive cells in the absence of KLF5 upon prolonged ATR inhibition (1 µM or 2 µM for 24 h) might be due to cell death-related processes such as apoptosis and caspase activation rather than ATRi-induced replication stress and DNA damage. To address this possibility, we firstly assessed apoptosis by using Annexin V staining and observed that 24-h treatment with different concentrations of ATRi did not induce apoptosis in WT and *KLF5* KO U2-OS or RPE-1 cells (Supplementary Fig. 3d, e). To evaluate whether the high γH2AX signal was a result of caspase activation prior to actual cellular death, we treated cells with ATRi in combination with the pan-caspase inhibitor ZVAD-FMK and assessed percentage of γH2AX-positive cells. As shown in Supplementary Fig. 3f, g, ZVAD treatment following ATRi treatment did not significantly affect the percentage of γH2AX positive cells in either U2-OS or RPE-1 backgrounds, suggesting that γH2AX induction reflected DNA damage and replication stress induced by ATRi and not cell death-associated processes. Next, we performed RNA-sequencing (RNA-seq) to compare transcriptional profiles of *KLF5* KO and parental U2-OS cells (Supplementary Data 2). Pathway analysis identified several pathways that were differentially regulated upon KLF5 loss, among them "DNA replication", "mitotic cell cycle", "DNA repair", and "DNA recombination" (Supplementary Fig. 3h), further explaining the observed phenotypes. We thus conclude that the marked increase in S phase DNA damage upon ATRi in *KLF5* KO cells, leads to cells progressing through S phase and entering mitosis with damaged and under-replicated DNA. This ultimately results in increased genome instability as observed by increased micronuclei formation upon ATRi treatment (Fig. 3g) and thus ATRi sensitivity.

## KLF5 loss increases transcription-replication conflicts and DNA-RNA hybrids

Based on its known functions as a transcription factor, we hypothesised that aberrant RNA polymerase II (RNAPII) transcription following KLF5 loss might lead to increased transcription-replication conflicts (TRCs), R-loops and DNA damage. To test this, we performed proximity ligation assays (PLAs) to measure collisions between the transcription and replication machineries by using antibodies against the elongating form of RNAPII phosphorylated on Ser-2 and the DNA replication protein, PCNA, following ATRi or CHK1i treatment of cells. Compared to WT U2-OS cells, *KLF5* KO cells displayed elevated levels of TRCs even in untreated conditions, and these TRC levels increased further above WT levels upon either ATRi or CHK1i treatment (Fig. 4a, b), and co-treating cells with ATRi and the RNAPII transcription inhibitor, DRB (5,6-dichloro-1-beta-ribo-fuanosyl benzimidazole) reduced TRCs in both WT and *KLF5* KO cells (Fig. 4a, b). These results implied that in untreated conditions, KLF5-depleted cells experience high levels of TRCs which is exacerbated upon ATRi or CHK1i treatment.

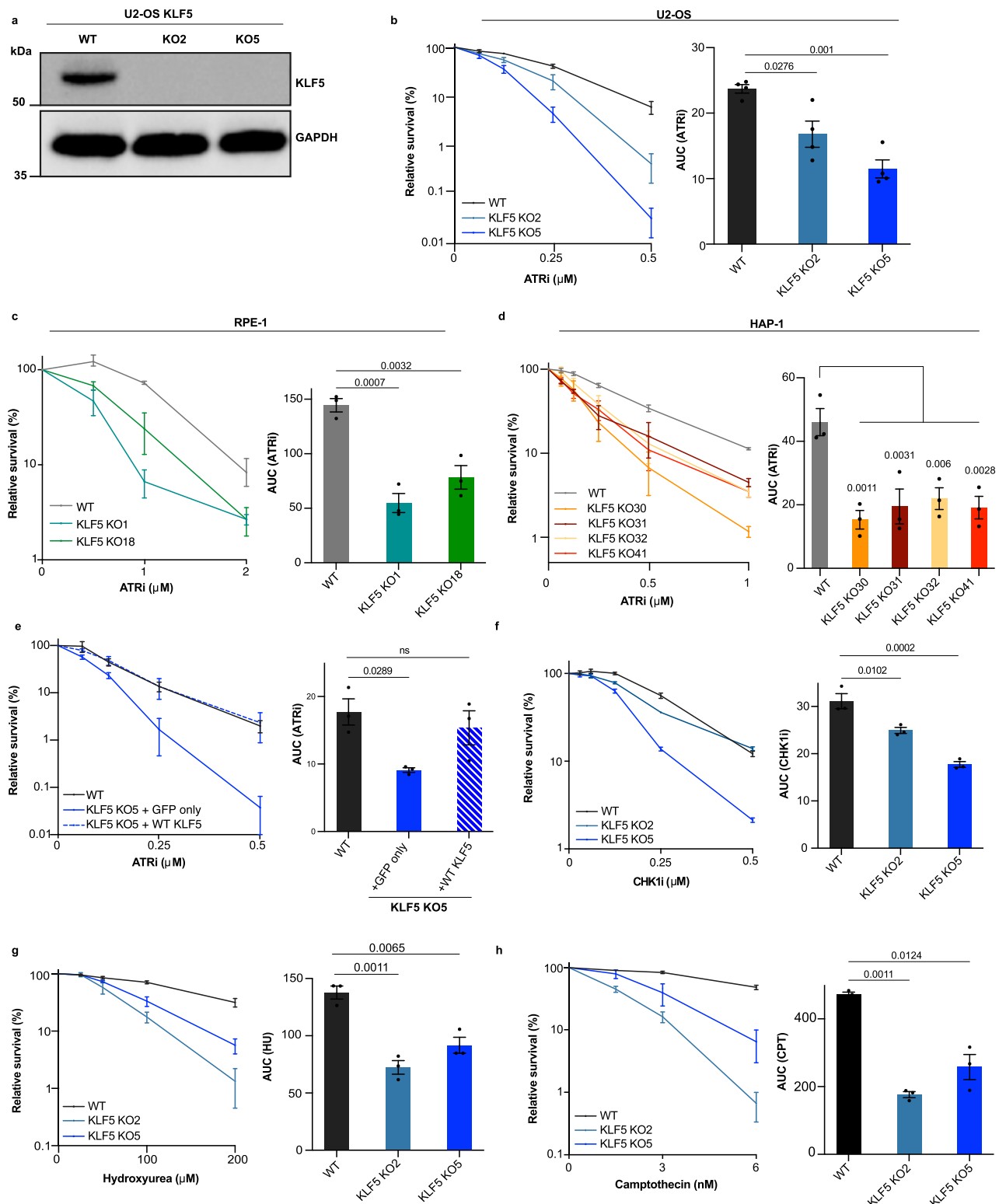

**Fig. 2 | Loss of KLF5 hypersensitizes cells to replication stress. a** Western blot analysis of KLF5 protein levels in U2-OS cells following CRISPR-mediated gene knockout (KO). GAPDH was used as loading control. Representative of at least 5 independent experiments. **b**, **c** Clonogenic survivals of WT and *KLF5* KO U2-OS (**b**), and RPE-1 (**c**) cells treated with ATRi. **d** alamarBlue cell viability assay of WT and *KLF5* KO HAP-1 cells upon treatment with ATRi, (biological *n* = 3). **e** Clonogenic survivals of WT and *KLF5* KO U2-OS cells, complemented with GFP-only or with WT KLF5-GFP, treated with ATRi (AZD6738). **f**–**h** Clonogenic survivals of WT and *KLF5* KO U2-OS cells treated with CHK1i (**e**), hydroxyurea (**f**), or camptothecin (**g**). Error bars represent means ± SEM from at least three biologically independent experiments. Data represented as AUCs (area under the curves). Statistical analyses were performed using a one-way ANOVA test with multiple comparisons. ns not significant. Source data are provided as a Source Data file.

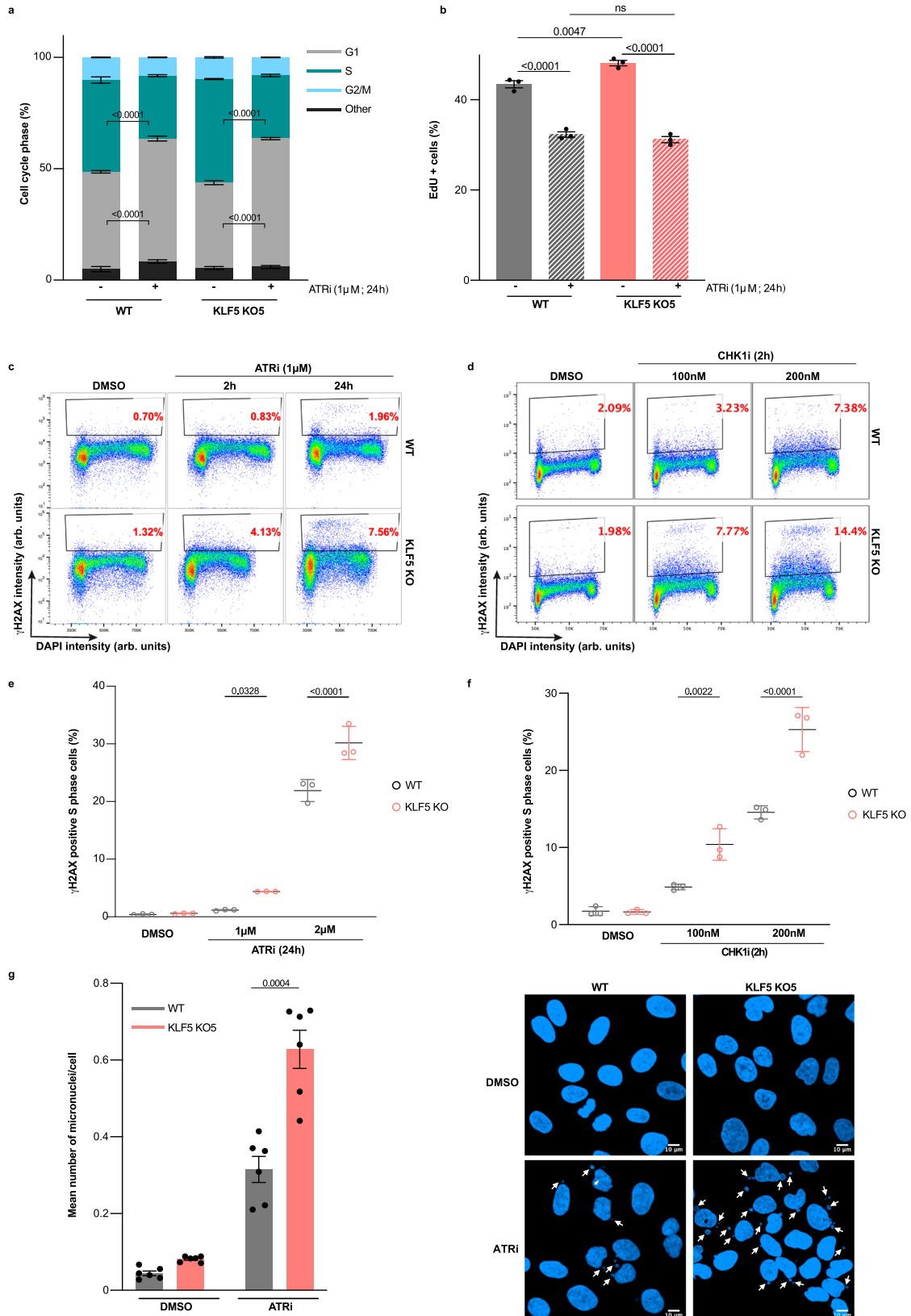

One source of TRCs is abnormal accumulation of R-loops, a three-stranded transcriptional by-product structure caused by the annealing of newly synthesised RNA with its complementary DNA, thus forming a DNA-RNA hybrid, and a displaced single-stranded DNA region[39–41]. This prompted us to determine the effect of KLF5 loss on R-loop formation. Initially, we exploited a previously established U2-OS cell line stably expressing green fluorescent protein (GFP)-tagged catalytically inactive RNaseH1 protein mutant (D210N) to detect R-loops[42]. After treating these cells with control siRNA or siRNA to deplete KLF5, they were treated with ATRi or CHK1i and monitored for the chromatin-bound fraction of GFP-RNH1(D210N) following pre-extraction to remove non-chromatin bound GFP-RNH1(D210N). This assay indicated a significant

**Fig. 3 | Loss of KLF5 induces DNA damage and genomic instability upon ATRi.** **a** Bar graphs showing percentages of cells in each cell cycle phase based on EdU and DAPI staining of WT and *KLF5* KO U2-OS cells treated with DMSO or ATRi (biological *n* = 3). **b** Bar graphs showing percentages of EdU positive WT and *KLF5* KO U2-OS cells following ATRi treatment. Cells were labelled with 10 μM EdU for 30 min prior to fixation. Data are shown as means ± SEM; biological *n* = 3. **c** FACS plots showing the percentages of γH2AX positive cells in WT and *KLF5* KO U2-OS cells untreated or following ATRi treatment at indicated timepoints. **d** As in (**c**), except cells were treated with CHK1i. Percentages of γH2AX positive cells are shown in red.

**e** Percentages of γH2AX positive S phase cells following ATRi (1 μM or 2 μM for 24 h) treatment (biological *n* = 3). **f** Percentages of γH2AX positive S phase cells following CHK1i (100 nM or 200 nM for 2 h) treatment (biological *n* = 3). **g** Left, mean numbers of micronuclei per cell following 24 h treatment with DMSO or 1 μM ATRi in WT and *KLF5* KO U2-OS cells (biological *n* = 6). Right, representative images showing micronuclei indicated with white arrows, scale bars = 10 μm. All data are represented as means ± SEM. Statistical analyses were performed using two-way ANOVA test (**a**), unpaired two-sided Student's *t*-test (**b**, **g**) and one-way ANOVA test with multiple comparisons (**e**, **f**). Source data are provided as a Source Data file.

increase in accumulation of R-loops upon ATRi or CHK1i in KLF5-depleted cells when compared to control WT cells (Supplementary Fig. 4a–c).

To corroborate these observations, we decided to map the global distributions of R-loops in WT and KLF5 depleted cells upon ATRi treatment. Thus, after treating *KLF5* KO and WT U2-OS cells with or without ATRi, we performed DNA:RNA immunoprecipitation (DRIP) assays by using the monoclonal S9.6 antibody, which binds DNA-RNA hybrids with high affinity[43], followed by high-throughput sequencing (DRIP-seq). Ensuing analysis revealed significantly higher levels of DNA-RNA hybrids in *KLF5* KO cells upon ATRi treatment compared to WT cells (Fig. 4c; note that as shown in Supplementary Fig. 4e, we observed a reduction of S9.6 signals in RNaseH treated samples compared to untreated samples). Moreover, mapping DRIP signals to genomic loci revealed that, out of over 20,000 sites with ATRi-induced DNA-RNA hybrids, ~20% were common between WT and *KLF5* KO cells, ~28% appeared unique to WT U2-OS cells and over 50% were specific to *KLF5* KO cells (Fig. 4d and Supplementary Fig. 4f). Notably, the sites of DNA-RNA hybrids that we detected were predominantly enriched at transcription start sites (TSS) as compared to gene bodies and transcription end sites (TES; Fig. 4e), with *KLF5* KO cells showing higher levels of R-loops at such sites than control cells. These data were consistent with a previous report showing that TSS are hotspots for R-loops[44]. To gain further insight and evaluate whether transcriptional dysregulation caused by KLF5 at specific genes is associated with abnormal formation of R-loops, we thoroughly compared our DRIP-seq and RNA-seq data and found a statistically significant overlap between genes with dysregulated expression and genes with increased R-loops (Supplementary Fig. 4g), suggesting that transcriptional dysregulation caused by KLF5 at specific genes may be associated with abnormal formation of R-loops specifically at those genes. Taken together, our results suggested that KLF5 loss exacerbates R-loop levels at various genomic sites, with many of these corresponding to TSS regions, at least some of which are already prone to forming R-loops under normal settings.

### KLF5 protects cells from transcription-dependent replication stress

Aberrant formation and accumulation of TRCs and R-loops have harmful consequences for genomic stability, mainly due to fostering replication-fork stalling and DNA DSB generation[12,13]. Under conditions of replication stress, ATR is activated to promote fork protection and stability, and through its downstream effector CHK1, it works globally to inhibit new replication-origin firing[45]. However, in the absence of fork protection, replication stress leads to generation of extensive single-stranded DNA that causes RPA exhaustion, followed by fork breakage and DSB formation[46]. We therefore investigated whether KLF5 loss increases replication stress and replication catastrophe upon ATR or CHK1 inhibition. By using high-content microscopy for quantitative image-based cytometry (QIBC) and flow cytometry, we assessed levels of replication stress by measuring the percentages of cells positive for chromatinized RPA32 staining, and replication catastrophe (RC) by measuring the percentages of RPA32 and γH2AX dual-positive cells following ATR inhibition. This revealed that both U2-OS and RPE-1 *KLF5* KO cells experienced higher levels of replication stress and RC

when compared to WT cells upon relatively short-term (2 h) or long-term (24 h) exposure to ATRi (Fig. 5a and Supplementary Fig. 5a–c) in a dose dependent manner. Similarly, when using CHK1i, we observed that *KLF5* KO cells had higher levels of replication stress and RC (Fig. 5b). As a complementary approach, we used western blot analysis to assess the activation of markers of replication stress and DSB formation upon ATRi in both WT and *KLF5* KO U2-OS cells. In line with the QIBC and flow cytometry results, western blot analysis confirmed that *KLF5* KO cells experienced higher levels of replication stress and DSB formation upon ATRi when compared to WT cells, as measured by the levels of RPA32 phosphorylation on Ser-4/8 and γH2AX (Fig. 5c), and this occurred in a dose-dependent manner. Collectively, these findings indicated that loss of KLF5 increases replication stress and DNA damage, particularly in the presence of ATRi or CHK1i. Importantly, we did not observe a difference in the activation of ATM, as shown by its auto-phosphorylation on Ser-1981 and the phosphorylation of its downstream effector CHK2 on Thr-68 (Fig. 5c), which means that the KLF5 loss hypersensitizes cells to ATRi but does not markedly affect ATM pathway activation.

TRCs and excessive formation of DNA-RNA hybrids present obstacles to replication forks, leading to fork stalling and collapse, and are thus considered a key source of genomic instability[12,13]. To explore whether the observed phenotypes of KLF5 loss on replication stress, replication catastrophe and sensitivity to ATRi are due to the accumulation of TRCs and R-loops, we assessed the impact of overexpressing WT RNaseH1, which resolves DNA-RNA hybrids, on replication stress, RC and cell survival following ATRi. Thus, we observed that overexpressing WT RNaseH1 decreased the proportion of KLF5-depleted cells experiencing replication catastrophe (Fig. 5d) and enhanced survival of KLF5-depleted cells upon ATRi (Fig. 5e). Collectively, these data indicated that aberrant transcription and its side-products— TRCs and DNA-RNA hybrid formation—that arise upon depleting KLF5 are likely directly responsible for ATRi-induced replication stress and cell death.

### KLF5 regulates chromatin recruitment of BRD4

A recent study showed that KLF5 directs histone acetyltransferase CBP/EP300 to chromatin to increase acetylated histone H3 lysine 27 (H3K27ac), thus regulating recruitment of bromodomain protein BRD4 to chromatin[47]. Moreover, several reports have implicated BRD4 in preventing the accumulation of TRCs and R-loops[48–50]. Prompted by these findings, we hypothesised that the phenotypes we observed upon KLF5 loss might be through disrupting BRD4 cellular localisation. To address this, we firstly performed chromatin fractionations in WT and *KLF5* KO U2-OS and RPE-1 cells and assessed levels of chromatin-bound BRD4. We observed that, while both long and short isoforms of BRD4 were present in the chromatin-bound fraction of WT cells, levels of chromatin-bound BRD4 long isoform were substantially reduced in U2-OS *KLF5* KO cells (Fig. 6a; note that the middle band in the BRD4 blot is a cross-reacting protein species, as established in Fig. 6d). Reduced chromatin-bound BRD4 was also observed in *KLF5* KO RPE-1 cells (Supplementary Fig. 6a; in this setting, both long and short isoforms were affected). These effects were not due to KLF5 reducing overall levels of histone proteins (Fig. 6a and Supplementary Fig. 6a) or BRD4 levels, which were comparable in the whole cell extracts of WT

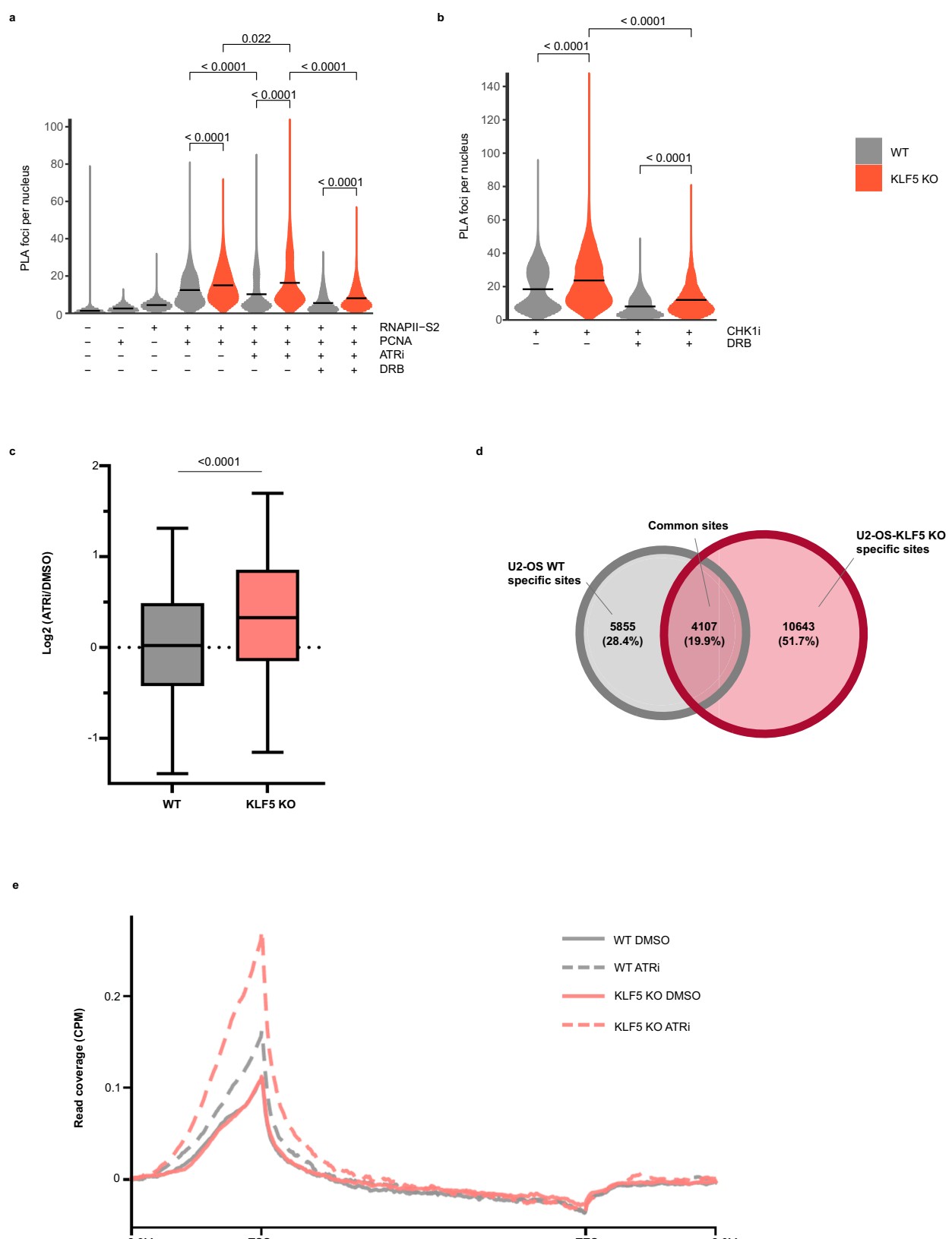

and KLF5 depleted U2-OS and RPE-1 cells (Fig. 6a and Supplementary Fig. 6a). Importantly, the long isoform of BRD4 contains the C terminal domain, and is the main isoform accountable for regulating RNAPII elongation through its ability to activate P-TEFb (positive transcription elongation factor b) to release paused RNAPII[51] and thus prevent transcription-associated DNA damage[48,49].

As an approach complementary to chromatin fractionation, we performed Cleavage Under Targets and Tagmentation (CUT&Tag) for high-resolution genome-profiling of BRD4 in both WT and *KLF5* KO U2-OS cells. Firstly, we assessed binding sites of KLF5 in WT and *KLF5* KO U2-OS cells by using a KLF5 antibody (Supplementary Fig. 6b). Next, we used an antibody against BRD4 and assessed its binding sites in WT and

**Fig. 4 | KLF5 loss increases transcription-replication conflicts and DNA-RNA hybrids. a** Dot plots showing quantification of PLA foci that represent interaction between RNAPII P-S2 antibody and PCNA in WT and *KLF5* KO U2-OS cells. No antibody or single antibody staining was used as negative controls. Where indicated, cells were treated with 1 µM ATRi for 24 h or combined with RNAPIIi (DRB,100 µM) 2 h prior to fixation; data are represented from three independent biological replicates. Mean numbers of PLA foci for each replicate were used for overall mean calculations and statistical analyses using two-sided Mann–Whitney test. **b** As in a except cells were treated with CHK1i (LY2603618, 200 nM) for 2 h or combined with DRB. Mean numbers of PLA foci for each replicate were used for overall mean calculations and statistical analyses using two-sided Mann–Whitney

test. **c** Boxplots showing the $\log_2$ ratio of DRIP signal between ATRi-treated and UT conditions for WT or *KLF5* KO U2-OS cells. The box ends represent upper and lower quartiles, the centre line represents median, and the whiskers represent 10th and the 90th percentiles. Statistical analyses were perform using two-sided, unpaired Wilcoxon test (biological *n* = 3). **d** Venn diagram showing the number of genomic sites that show enrichment of DRIP signal specific to WT (grey) or *KLF5* KO (red) cells, as well as common sites. **e** Metagene plot of the distribution profile of S9.6 signals along all genes and flanking regions (±2 kb) in WT (grey) and *KLF5* KO (red) U2-OS cells treated with either DMSO or ATRi. Source data are provided as a Source Data file.

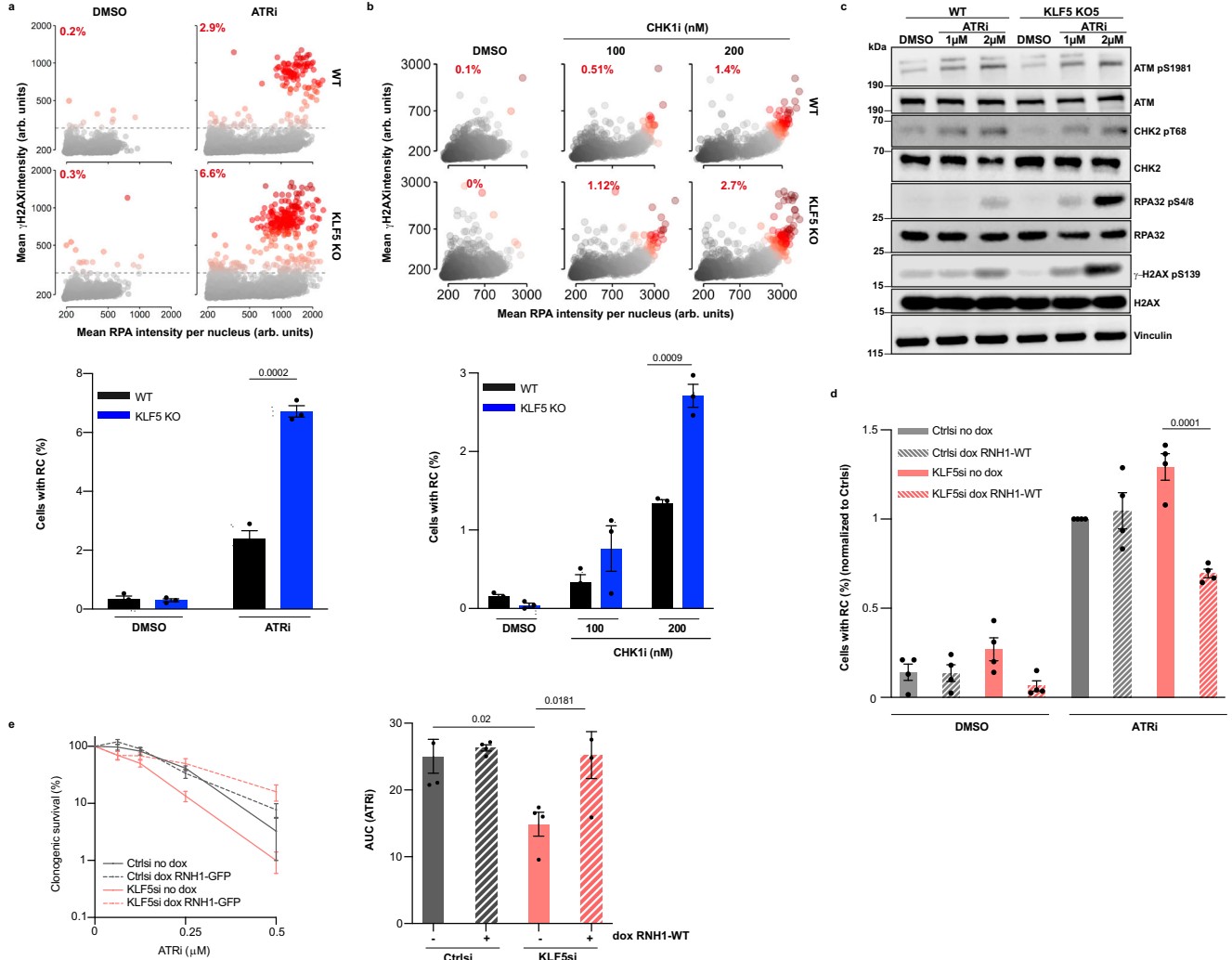

**Fig. 5 | KLF5 protects cells from transcription-dependent replication stress. a Top**, QIBC analysis of WT and *KLF5* KO U2-OS cells DMSO or 1 µM ATRi treated. Percentages of cells in replication catastrophe (RC); hyper-positive for RPA and γH2AX dual-positive, are provided in red. **Bottom**, quantification of cells with RC. **b** As in a, except of cells treated with CHK1i at the indicated concentrations for 2 h. Data are represented as means ± SEM from three independent biological replicates. Statistical analyses were performed using an unpaired two-sided *t*-test. **c** Western blot analysis of replication stress and DSB formation markers. WT and *KLF5* KO U2-OS cells were treated for 24 h with 1 µM or 2 µM ATRi prior to lysis. The samples derive from the same experiment but different gels, one gel for pATM, Vinculin,

γH2AX, pRPA32, another gel for ATM, H2AX, RPA32, another gel for pCHK2, and another gel for CHK2. Representative of at least 3 independent experiments. **d** Quantification of cells in RC upon overexpressing WT RNaseH1 after transfection with Ctrlsi or KLF5si and ATRi treatment. Error bars = mean ± SEM (biological *n* = 4). Statistical analyses were performed using one-way ANOVA with multiple comparisons. **e** Clonogenic survivals of U2-OS with control or KLF5 siRNA depletion with or without doxycycline induction of WT RNH1-GFP and treated with ATRi. Error bars = mean ± SEM (biological *n* = 4). Data represented as AUCs. Statistical analyses were performed using one-way ANOVA with multiple comparisons. Source data are provided as a Source Data file.

*KLF5* KO U2-OS cells. Analysis of the CUT&Tag assay data identified total BRD4 binding sites in WT U2-OS cells, and KLF5-regulated BRD4 binding sites, which were present in WT cells but abrogated upon *KLF5* KO (see Fig. 6b and Supplementary Fig. 6c for representative genomic regions showing KLF5 binding in WT and KLF5 KO cells, and reduced

BRD4 binding following KLF5 loss; and Fig. 6c showing metagene and heatmap for a broader analysis of identified BRD4 binding sites in U2-OS cells).

These results supported the notion that KLF5 and BRD4 work in a largely epistatic manner regarding ATRi sensitivity. In accord with this,

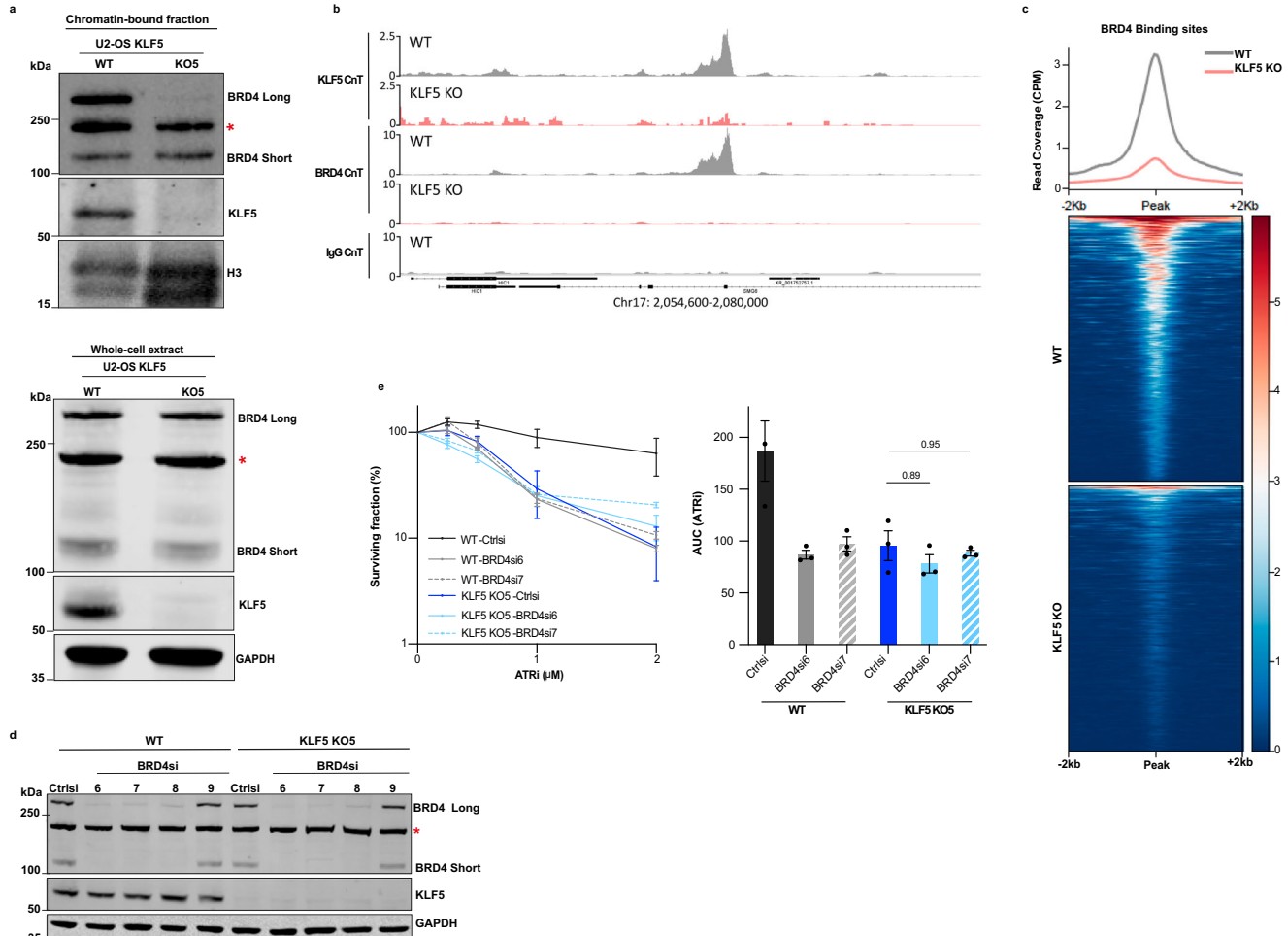

**Fig. 6 | KLF5 regulates BRD4 recruitment to chromatin. a** Western blot analysis of chromatin-bound BRD4 long and short isoforms in WT and *KLF5* KO U2-OS cells (Top). Histone H3 was used as marker for chromatin-bound fraction. And whole-cell extracts (bottom). *- cross-reacting protein species. Representative of at least 3 independent experiments. **b** Genome browser views of KLF5 CUT&Tag (CnT) (upper two panels), BRD4 CnT (middle two panels), and the negative control using IgG antibody (lower panel) in WT (grey peaks) and *KLF5* KO (red peaks) U2-OS cells. **c** Metagene (upper panel) and Heat map (lower panel) demonstrating changes in BRD4 chromatin binding upon KLF5 loss. BRD4 binding sites were called based on WT relative to *KLF5* KO cells, (*n* = 3). **d** Western blot analysis showing BRD4 levels upon siRNA depletion in WT and *KLF5* KO U2-OS cells. GAPDH was used as a loading control. *- cross-reacting protein species. L: Long, S: Short forms of BRD4 protein. Representative of at least 3 independent experiments. **e** Clonogenic survivals of WT and *KLF5* KO5 U2-OS ± BRD4 siRNA depletion treated with ATRi. Error bars = means ± SEM (biological *n* = 3). Data represented as AUCs. Statistical analyses were performed by one-way ANOVA with multiple comparisons. Source data are provided as a Source Data file.

we found that while siRNA depletion of BRD4 in WT sensitised them to ATRi, its depletion in the *KLF5* KO background did not further increase the sensitivity of cells to ATRi (Fig. 6d, e). Given BRD4's known function in regulating the activity of RNAPII[52], we thus suggest that KLF5 regulates chromatin binding of BRD4, thereby affecting promoter-proximal pause release of RNAPII and transcription elongation to protect cells from TRCs, DNA-RNA hybrids, replication stress and genomic instability.

To examine more directly whether the increase in R-loops caused by KLF5 depletion was preferentially at genes where BRD4 recruitment was dysregulated, we compared between R-loop levels obtained from DRIP-seq and BRD4 binding sites obtained from CUT&Tag. However, as BRD4 binds to enhancer regions[52] that are not generally located within genes and instead regulate the expression of distant genes, we identified the genes located closest to the BRD4 binding sites and assessed R-loop levels at those sites. Such analysis revealed a significant increase in R-loop levels at genes located closest to BRD4 binding sites in the *KLF5* KO cells (Supplementary Fig. 6d), however this did not appear to be greater than the genome-wide impact that we observed in Fig. 4e.

### KLF5 loss exacerbates R-loop-dependent DNA damage and is lethal for ARID1A null cells

Finally, we sought to decipher the molecular basis underlying the synthetic lethality between ARID1A and KLF5. Prompted by our findings showing that KLF5 regulates chromatin binding of BRD4, we sought to assess whether KLF5 is required for chromatin recruitment of BRD4 in *ARID1A* KO cells. To do so, we performed chromatin fractionation in WT, *ARID1A* KO and *ARID1A* KO RPE-1 cells transfected with KLF5 siRNA, and tested for effects on the BRD4 chromatin-bound fraction. We found that KLF5 was required for effective recruitment of BRD4 to chromatin in *ARID1A* KO cells (Supplementary Fig. 7a). These results are in line with previous observations showing that BRD4 inhibition is synthetic lethal for ARID1A mutant cells[53]. Accordingly, and supported by ARID1A's function in modulating R-loops[54–56], we hypothesise that upon loss of both KLF5 and ARID1A, cells experience higher levels of R-loops, thereby compromising their proliferation.

To address this hypothesis, we firstly carried out RNA-seq in *ARID1A* KO cells (Supplementary Data 2) and assessed the transcriptional profiles of a subset of R-loop regulators in *ARID1A* KO and *KLF5*

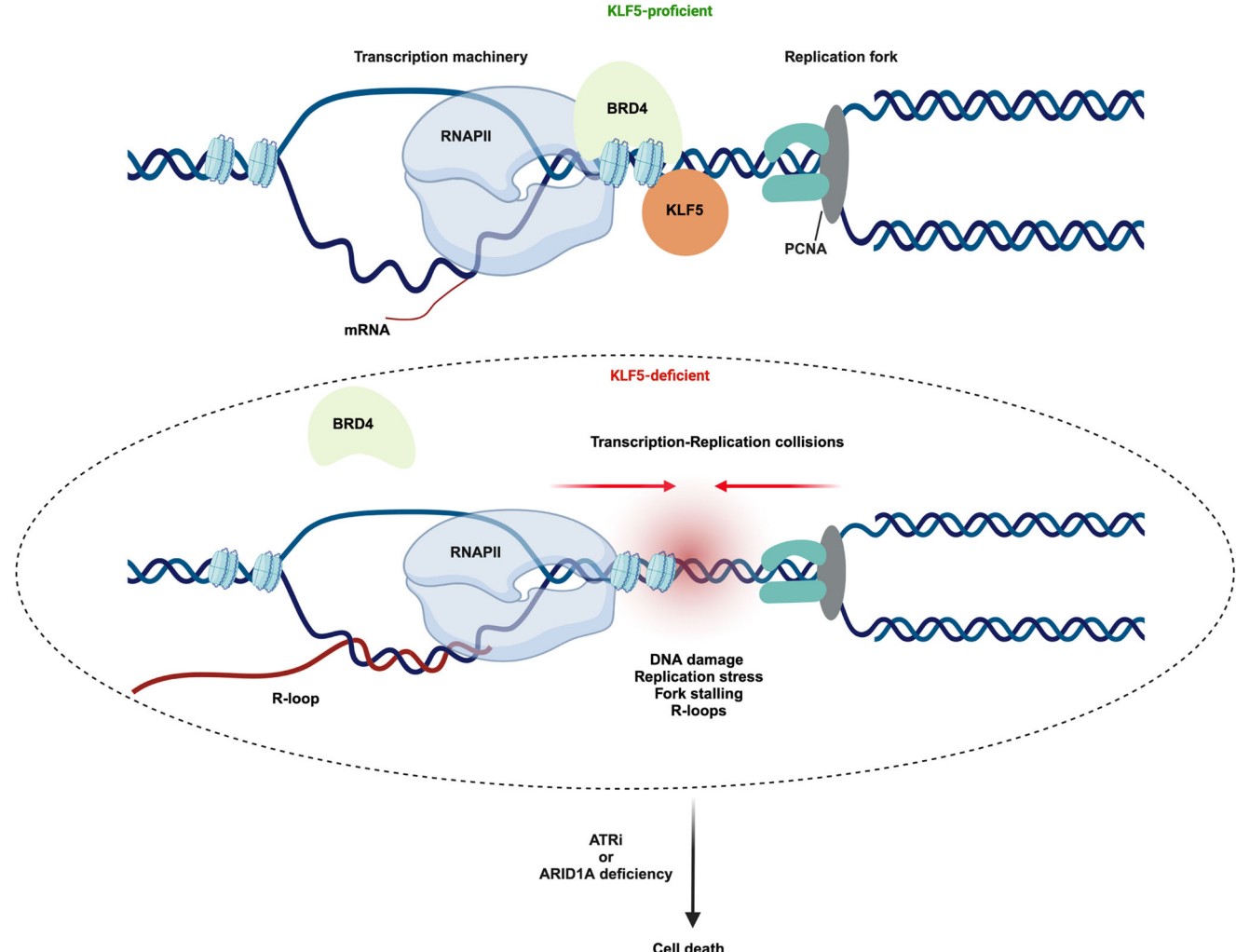

**Fig. 7 | A proposed model for KLF5's role in protecting cells against transcription-dependent replication stress.** Top: KLF5 binds DNA and facilitates BRD4 recruitment to chromatin, thus regulating RNAPII promoter-proximal pausing and transcription elongation in a manner that prevents transcription-associated DNA replication stress. Bottom: Upon KLF5 loss, the recruitment of BRD4 to chromatin is disrupted, causing increased transcription-replication conflicts and downstream events such as DNA replication stress, DNA damage, and increased R-loop accumulation, which when combined with ATR inhibition or with ARID1A deficiency, causes genomic instability and cell death. Created in BioRender. Diab, S. (2025) https://BioRender.com/r68r668.

KO in comparison to WT U2-OS cells. Such analysis revealed that both ARID1A and KLF5 are involved in modulating the expression of regulators of R-loops such as DHX9, BRCA1, FANCD2 and TREX1 (Supplementary Fig. 7b). Secondly, we co-depleted ARID1A and KLF5 and assessed levels of the DNA damage marker, γH2AX in the absence or presence of WT RNaseH1. As shown in Supplementary Fig. 7c, co-depletion of ARID1A and KLF5 lead to increased γH2AX signal (2.4-fold increase) when compared to control or the single depletion of either protein, and the increase in γH2AX was diminished upon WT RNaseH1 overexpression (2.1-fold decrease) (Supplementary Fig. 7c). These results therefore suggested that both ARID1A and KLF5 are involved in the homeostasis of R-loops, and their combined depletion perturbs R-loops sufficiently to cause accumulation of DNA-damage and ensuing cell death and/or slow cell proliferation.

## Discussion

Our study has revealed that KLF5 loss hypersensitizes WT cells to ATRi, and is synthetic lethal for ARID1A mutant cells, in a manner associated with higher levels of transcription-replication conflicts and R-loop accumulation. Hence, we propose a model highlighting a new function of KLF5 in protecting cells from transcription-associated replication stress, DNA damage and genome instability through controlling the levels of R-loops by regulating the chromatin recruitment of BRD4 and the transcription of R-loop regulators. Accordingly, in the absence of KLF5, chromatin-binding of BRD4 (mainly the long isoform) is disrupted, leading to an increase in TRCs, R-loops and associated replication stress. Consequently, when loss of KLF5 is combined with either ATR inhibition or ARID1A deficiency, this compromises cell proliferation (Fig. 7).

Functionally, we have found that KLF5 safeguards cells from transcription-induced replication stress, at least in part through regulating chromatin recruitment of the transcriptional coactivator, BRD4. In line with data from the Human Protein Atlas showing that KLF5 is universally expressed across human tissues, and our findings showing that KLF5 promotes chromatin recruitment of BRD4 in multiple cells lines; in the osteosarcoma cell line, U2-OS, the untransformed human retinal pigment epithelial-1 (RPE-1) cell line, and in the lung squamous cancer cell line, HARA[47], these observations therefore suggest that KLF5 is widely if not universally required for recruiting BRD4, rather than cell-line/type specific. Of note, data from DepMap indicate that various cancer cell lines are particularly dependent on KLF5 expression[37], thereby highlighting how targeting KLF5 in combination with ATR inhibitors could provide an opportunity for treating KLF5-dependent cancers.

KLF5 belongs to the KLF transcription factor family, which in human cells consists of 17 members. Despite the similarity in their overall protein structure, whether other KLFs are involved in regulating BRD4 recruitment to chromatin has not been experimentally demonstrated. However, our CRISPR screen data suggest that regulation of BRD4 and impacts on cellular responses to ATRi are largely specific to KLF5—that is, if other KLFs are redundantly involved in such functions, this would have obscured the impacts of KLF5 deficiency that we have clearly observed. Furthermore, based on our analysis of DepMap data, targeting KLF4 in *ARID1A*-deficient cancer cells does not appear to influence the fitness of *ARID1A* mutant cell lines, further highlighting the specificity of the effect of loss of KLF5 on *ARID1A*-deficient cells.

Our study has revealed that KLF5 and BRD4 operate in an epistatic manner to combat transcription-driven replication stress—observations that are consistent with the broad role of BRD4 in regulating the release of promoter-proximal paused RNAPII and transcription elongation, through interaction with P-TEFb, and preventing R-loop formation[48–50,57]. Accordingly, we found that overexpressing WT RNaseH1 rescued the ATRi hypersensitivity of *KLF5* KO cells, thereby providing further support to the concept that normal spatiotemporal organisation between the transcription and replication machineries is vital to maintain genome integrity and cell survival, and that ATR activity is required for resolving transcription-replication conflicts when they arise[58]. Paradoxically, our PLA assay showed that in WT cells, the levels of TRCs are reduced upon ATRi treatment, an observation that does not support ATR functions in resolving TRCs and preventing the accumulation of R-loops[59,60]. However, such reduction in TRCs in WT cells was observed when cells were treated with ATRi for 24 h. Therefore, we speculate that during the early timepoints, there might be an increase in TRCs upon ATRi in WT cells; however, proficient R-loop regulation in WT cells might result in a reduction at the end point, 24 h post ATRi treatment. This assumption is further supported by the observed increase in R-loops (measured by DRIP-seq and RNaseH1 D210N) in both WT and *KLF5* KO cells upon ATRi, yet the increase in *KLF5* KO cells is far more pronounced. As R-loops are considered a source of TRCs[41], this correlates with the elevation of TRCs in *KLF5* KO cells. Nevertheless, precisely how accumulation of R-loops interferes with the replication machinery and causes replication stress remains to be further explored.

Our data provide several lines of evidence supporting the idea that the synthetic lethality between KLF5 and ARID1A is also due to dysregulating R-loop homeostasis. Firstly, we show that KLF5 is required for effective chromatin recruitment of BRD4, which is required for preventing R-loop accumulation[48–50], in *ARID1A*-deficient cells. Secondly, we show that both ARID1A and KLF5 regulate the transcription of several regulators of R-loops. Finally, we show that the concurrent depletion of KLF5 and ARID1A increases R-loop-dependent DNA damage. Evidence supporting these observations includes cells lacking ARID1A being dependent on BRD4 for proliferation, and that BRD4 inactivation using JQ1 inhibitors is toxic to *ARID1A*-mutant cells[53], leading us to speculate that KLF5 loss mimics BRD4 inhibition and behaves as a synthetic lethal target for ARID1A.

Bromodomain inhibitors, such as JQ1, and proteolysis targeting chimera (PROTAC) molecules targeting BRD4 for degradation, such as dBET6, have been developed and are in clinical trials[61]. However, due to structural similarities between different bromodomain containing proteins, these compounds have been reported to exert various effects through affecting other bromodomain-containing proteins, such as BRD2 and BRD3. Our data provide an alternative potential route for targeting chromatin binding of BRD4 through modulating KLF5, in a manner which synergises with ATR inhibition. As ARID1A is frequently mutated in various types of human cancer—including ovarian, gastric, pancreatic, and colorectal cancers[25–27]—and its mutations result predominantly in loss of function, our study may have important translational implications through it suggesting KLF5 as a potential therapeutic target for *ARID1A*-deficient cancers.

## Methods

### Cell lines

Cell lines used in this study were routinely tested for mycoplasma contamination[62]. U2-OS, CAL-51 and MCF-7 cells were cultured in Dulbecco's modified Eagle's medium (DMEM, Gibco), RPE-1 in Dulbecco's Modified Eagle Medium: Nutrient Mixture Ham's F-12 (DMEM/F-12, Gibco), and HAP-1 in Iscove's modified Dulbecco's medium (Thermo Fisher Scientific). All cell lines were grown at 37 °C and 5% CO$_2$, and cultured in media supplemented with 10% FBS, 2 mM L-glutamine, 100 units/ml penicillin, and 100 μg/ml streptomycin. 10 μg/ml blasticidin (Sigma-Aldrich) was used to select for Cas9 expressing cells. Cells were additionally cultured in the presence of 1 μg/ml of puromycin during selection of the infected cells in the CRISPR-Cas9 screens. U2-OS T-Rex GFP-RNase H1(D210N) or GFP-RNase H1-WT cells were a kind gift from Pavel Janscak[42], MCF-7 WT and *ARID1A* KO cells were a kind gift from Jason S. Carroll[53]. To generate *ARID1A* and *KLF5* knockouts, Cas9-expressing U2-OS or RPE-1 cells were transiently transfected with single guide RNAs (gRNAs), and HAP-1-Cas9 cells were transduced with sgKLF5. Single cell clones were tested for successful and complete gene knockouts via western blotting. All KO clones used in this study were genotype validated by genomic DNA extraction and PCR amplification of the sgRNA-targeted loci (gRNAs and primers used can be found in Supplementary Table 2) followed by either TIDE[63] or TOPO-cloning (Thermofisher, K28002) and Sanger DNA sequencing.

### Compounds

ATR inhibitor (AZD6738), CHK1 inhibitor (LY2603618), KLF5 inhibitor (ML264) and Z-VAD(OH)-FMK (Caspase Inhibitor VI) were obtained from SelleckChem. DRB (5,6-dichloro-1-beta-ribo-fuanosyl benzimidazole), hydroxyurea, camptothecin and etoposide were obtained from Sigma-Aldrich.

### Clonogenic cell survival assays

U2-OS and RPE-1 cell lines were plated in triplicate in 6-well plates at a density of 500 cells/well. 16 h post seeding, media was replaced with media containing drug at the indicated doses. Cells were grown for 10–14 days until colonies formed. Colonies were stained using crystal violet, counted, and normalised to untreated conditions.

### Plasmid, viral transduction, and siRNA transfections

GFP only, GFP-KLF5 WT, or GFP-ARID1A WT plasmids were obtained from VectorBuilder. To generate inducible cell lines, lentiviruses were first generated in LentiX 293T cells by transfecting packaging plasmids psPAX2 (Addgene, #12260) and pMD2.G (Addgene, #12259) with the plasmid of interest using the transfection reagent TransIT-LT1 (Mirus Bio) according to the manufacturer's protocol. 72 h later, supernatant was collected and filtered through a 0.45 μm sterile Millex-GP filter unit (Merck). Lentivirus was stored at −80 °C. Viral transdcution was done by incubating viral particles with cells in the presence of 8 μg/ml polybrene. 24 h post transduction, cells were selected with puromycin. To induce expression of constructs, cells were incubated with 1 ng/ml doxycyclin for 24 h and then subjected for downstream applications. siRNA transfection was done with Lipofectamine RNAiMAX (Invitrogen) according to the manufacturer's protocol. siRNAs were obtained from Dharmacon and their sequences are listed in Supplementary Table 3.

### CRISPR–Cas9 screen combined with ATRi

WT, *ARID1A* KO2, or *ARID1A* KO24 U2-OS cells stably expressing Cas9 were transduced at an MOI (multiplicity of infection) of 0.3 and 500× coverage of the genome wide Brunello library[64]. 48 h post transduction, cells were selected with puromycin (1 μg/ml) for 8 days. Cells were then divided into two populations and treated with either DMSO or IC50 ATRi (AZD6738) for 14 days. The IC50 ATRi dose of each cell line was predetermined based on pilot assays in non-transduced cells that were plated and passaged in a screen-matched format with various

ATRi concentrations. Cell pellets were collected on days 0 and 14 of treatment. Genomic DNA was then extracted from cell pellets using the QIAamp Blood Maxi Kit (Qiagen), DNA was amplified with Q5 Mastermix (New England Biolabs Ultra II) and i7 multiplexing barcoded primers. Following PCR product gel-purification, they were multiplexed with q-PCR NEBNext library quant kit (E7630). Products were sequenced on an Illumina NovaSeq 6000 system. Guide RNA enrichment analysis was performed with DrugZ to compare read counts between conditions[65].

### Flow cytometry
Cells were plated at a density of 500,000 cells/well in 6-well plates. The next day, cells were incubated with 10 μM EdU (5-ethynyl-2′-deoxyuridine; Sigma-Aldrich 900584) for 30 min prior to fixation. Cells were pre-extracted with PBS-T (PBS supplemented with 0.2% Triton X-100) for 10 min on ice, followed by 15 min fixation permeabilization with BD Cytofix/Cytoperm (BD bioscience) at room temperature. Antibody staining was performed in BD perm/wash buffer (BD biosciences) as required using the antibodies listed in Supplementary Table 1. The click reaction was subsequently performed as previously reported[66] before staining with 1 μg/ml DAPI in PBS containing 250 μg/ml RNase A. All samples were acquired using a Fortessa (BD biosciences) or A5 FACS Symphony (BD Biosciences) and analysed with FlowJo v.10.8.1.

### Western blot analyses
Cells were lysed in lysis buffer (50 mM Tris PH 7.5, 2% SDS, 10 mM N-ethylmaleimide) supplemented with protease inhibitor cocktail tablets (Roche), and phosphatase inhibitors (Sigma-Aldrich), and heated to 95 °C for 5 min. Protein samples were separated on 4–12% Bis-Tris NuPAGE gels, and transferred to a nitrocellulose membrane (Amersham) and immunoblotted using the antibodies at the indicated dilutions listed in Supplementary Table 1. Western blotting images were captured using ChemiDoc MP Imaging Sysytem (Bio-Rad).

### Immuno-fluorescence assays
Cells were seeded in 96-well plates (20,000 cells/well) or in 24-well plates (40,000 cells/well) 16 h before drug treatment. Cells were incubated with compounds for the indicated times. Cells were fixed in 4% PFA for 10 min at room temperature. For γH2AX and RPA32 staining, cells were pre-extracted in ice-cold CSK buffer (10 mM HEPES pH7.4, 300 mM sucrose, 100 mM NaCl, 3 mM MgCl$_2$) supplemented with 0.5% Triton X-100 for 10 min on ice prior to fixation. Cells were blocked with 5% BSA in 0.1% PBS−Tween (PBS-T) and incubated with primary antibodies at 4 °C overnight using antibodies at the indicated dilutions listed in Supplementary Table 1. Cells were washed three times in PBS-T and stained with Alexa Flour secondary antibodies for 1 h at room temperature in the dark, and then stained with DAPI (1 μg/ml). Plates were then imaged using an Opera Phenix spinning disc confocal microscope (PerkinElmer) at 40× magnification. Data were imported into Harmony image analysis software (PerkinElmer) for subsequent analyses.

### Chromatin fractionations
15 million cells were collected from the indicated cells lines, and chromatin fractionation was performed as previously described[67].

### Proximity ligation assays
20,000 cells/well were plated in 96-well imaging plates. 16 h later, cells were left untreated or treated with ATRi (1 μM, 24 h) or CHK1i (200 nM 2 h). Where indicated, cells were treated with 100 μM DRB for 2 h prior to fixation. Cells were pre-extracted with PBS supplemented with 0.2% Triton X-100 for 2 min on ice, and then fixed with 4% formaldehyde for 10 min. The PLA assay was then performed as previously described[21] with anti-PCNA and anti RNAPII-ser2 (see Supplementary Table 1). Following PLA, cells were counterstained with 1 μg/ml DAPI at room

temperature for 2 min and then washed with PBS×1. Plates were imaged with an Opera Phenix spinning disc confocal microscope (PerkinElmer) at 40× magnification. Data were imported into Harmony image analysis software (PerkinElmer) for subsequent analyses.

### RNH1(D210N)-GFP reporter assays
Detection of DNA-RNA hybrids (R-loops) with the RNH1(D210N)-GFP reporter assay was performed as previously reported[42]. Briefly, cells were transfected with the indicated siRNAs. 24 h post transfection, cells were plated in 24-well plates in the presence of doxycycline (1 ng/ml) to induce expression of GFP-RNase H1(D210N). 24 h later, cells were treated with DMSO, ATRi or CHK1i for the indicated times. Cells were pre-extracted with CSK buffer (10 mM HEPES−NaOH pH 7.4, 100 mM NaCl, 3 mM MgCl$_2$, 300 mM sucrose and 0.5% Triton X-100) for 5 min on ice before fixation in 4% PFA, and DNA was counterstained with DAPI (1 μg/ml). Image acquisition was performed on an Opera Phenix spinning disc confocal microscope (PerkinElmer) at 40× magnification. GFP nuclear intensity was analysed with Harmony image analysis.

### Chromatin profiling of KLF5 and BRD4 by CUT&Tag
Bulk CUT&Tag was performed as described previously[68], with some modifications. Dynabeads MyOne Streptavidin T1 beads (Thermo Fisher Scientific, 65601) were incubated with Concanavalin A (biotin conjugate, Sigma-Aldrich C2272) in ratio 10 mg/mL beads to 1.15 mg/mL Concanavalin A at 22 °C for 30 min at 400 rpm on a thermoshaker (Eppendorf). Volume sufficient for 10 μl beads per reaction was prepared. Beads were washed with 1 ml of binding buffer (20 mM HEPES pH 7.5, 10 mM KCl, 1 mM CaCl$_2$, 1 mM MnCl$_2$) and resuspended in a volume of binding buffer sufficient for 10 μl beads per reaction. U2-OS cells were harvested with StemPro Accutase (Thermo Fisher Scientific, A1110501) and $2 \times 10^6$ cells were fixed in 0.1% formaldehyde (Thermo Scientific, 28906) in 1 ml 1× PBS for 2 min at room temperature and quenched with 0.075 M glycine. Cells were centrifuged at 1300 × g 4 °C for 4 min and resuspended in wash buffer (20 mM HEPES pH 7.5, 150 mM KCl, 0.5 mM spermidine (Sigma, S0266), one tablet protease inhibitor (Roche cOmplete, EDTA-free, 11873580001)) at 500 cells/μl. 10 μl beads were incubated with 100 μl of cell suspension at 25 °C for 10 min at 600 rpm, washed twice with 100 μl wash buffer and resuspended in 50 μl of antibody buffer (2 mM EDTA, 0.1% BSA (Sigma A8577), 0.05% digitonin (Millipore, 300410) in wash buffer). All wash steps were performed on a DynaMag 2 magnet (Invitrogen). Primary antibody against KLF5, BRD4 and Normal Rabbit IgG were added to each sample at final concentration of (0.2 mg/mL) and incubated at 4 °C overnight at 600 rpm. Cells were washed twice with 100 μl dig-wash buffer (0.05% digitonin in wash buffer) and resuspended in 50 μl of dig-wash buffer. 0.5 μl of secondary antibody Guinea Pig anti-rabbit IgG (Antibodies-online, ABIN101961) was then added to each sample and incubated at 25 °C for 1 h at 600 rpm. Cells were then washed three times with 500 μl dig-wash buffer and resuspended in 50 μl dig-300 buffer (20 mM HEPES pH 7.5, 300 mM KCl, 0.5 mM spermidine, 0.01% digitonin, one tablet protease inhibitor). Mosaic-end adaptor loaded pA-Tn5 (made in house, as described previously[68], was then added to each sample in final 1:250 dilution to each sample and incubated at 25 °C for 1 h at 600 rpm. Cells were washed three times with 500 μl dig-300 buffer and resuspended in 300 μl tagmentation buffer (10 mM MgCl$_2$ in dig-300 buffer) and incubated at 37 °C 1 h at 600 rpm. Cells were then washed twice with 500 μl TAPS wash buffer (10 mM TAPS (Thermo Fisher Scientific J63268.AE) and resuspended in 100 μL of proteinase K solution (0.5 mg/mL Proteinase K (Thermo Fisher Scientific, EO0491), 0.5% SDS in 10 mM Tris-HCl pH 8.0). Samples were vortexed on full and incubated at 55 °C for 1 h at 800 rpm. DNA was extracted with DNA Clean & Concentrator 5 (Zymo Research, D4013) following the manufacturer's instructions, and DNA was eluted in 25 μl DNA elution buffer. Sequencing libraries were generated by mixing 25 μl NEBNext High Fidelity 2× PCR Master Mix (New England Biolabs, M0541), 2 μl of P5 indexing primer (10 μM), 2 μl of P7

indexing primer (10 μM) with 21 μl of purified DNA in a PCR reaction under the following conditions: 72 °C for 5 min, 98 °C for 30 sec followed by 10 cycles of 98 °C 10 sec, 63 °C for 10 sec, and a final extension at 72 °C for 1 min. Libraries were purified with Ampure XP beads (Beckman Coulter, A63882) by incubation with 0.4× ratio of beads for 15 mins, followed by supernatant incubation with 1.4× ratio of beads for 15 min. Beads were washed twice with 800 μl 80% ethanol and eluted in 25 μl of 10 mM Tris-HCl pH 8.0. Library size distribution was analysed on a Tapestation 2200 (Agilent Technologies), and library concentrations were quantified by NEBNext Library Quant Kit for Illumina (New England BioLabs, E7630S). Individual libraries were pooled for optimal read balancing and sequenced 59 bp paired end with NextSeq2000 P2 Reagents 100 Cycles v3 (Illumina, 20046811) on the NextSeq2000 platform (Illumina). Three technical replicates for each of three biological replicates were performed for each condition. Biological replicate 1 was performed and sequenced separately from the other replicates.

### DNA-RNA immunoprecipitation (DRIP)
DRIP assays were performed as previously described[69]. U2-OS cells were seeded in 15-cm plates, and 16 h later were either left untreated or treated with ATRi (AZD6736, 1 μM, 24 h). Cells were then washed with PBS, lysed in cytoplasmic lysis buffer (50 mM HEPES pH 7.9, 10 mM KCl, 1.5 mM MgCl$_2$, 0.34 M sucrose, 0.5% Triton, 10% glycerol, 1 mM DTT) for 10 min on ice and washed once in cytoplasmic lysis buffer. Cells were then scraped and lysed using nuclei lysis buffer (50 mM Tris pH 8.0, 5 mM EDTA pH 8.0, 1% SDS and 0.5 mg/ml proteinase K (Invitrogen, 25530049)) and incubated for 1 h at 55 °C for protein digestion. Genomic DNA was then precipitated by adding 10% volumes of 3 M sodium acetate pH 5.2 and 2.5× volumes of 100% ethanol and incubated at room temperature until DNA is clearly visibly precipitated. The DNA was spooled onto a P1000 pipette tip, washed once in 80% ethanol, air dried and resuspended in 300 μl 1× TE buffer. 50 μg of DNA was diluted into 300 μl of 1× TE and then sonicated for 13 cycles 10 sec on/30 sec off at 4 °C using Bioruptor Plus sonication device (Diagenode) and run on a 2% agarose gel to confirm fragmentation (see Supplementary Fig. 4d). 25 μg of fragmented DNA was then incubated with 50 units of RNase H (NEB, M0297L) for 3 h at 37 °C, topped up with an additional 50 units of RNase H for an extra 3 h. Next, 20 μg of each sample was immunoprecipitated overnight at 4 °C with 10 μg S9.6 antibody (Merck, MABE1095) conjugated to 100 μl Pierce dynabeads protein G magnetic beads (Thermo, 10004D). Beads were then washed once with 1× DRIP buffer, once with 1× DRIP buffer supplemented with 500 mM NaCl, and once in LiCl buffer (10 mM Tris pH 8.0, 250 mM LiCl, 1 mM EDTA pH 8.0, 1% NP-40) followed with two washes in 1× TE buffer. To elute DNA from beads, 100 μl genomic extraction buffer (1× TE buffer, 0.5% SDS, and 1 mg/ml proteinase K) was added, and beads were incubated for 30 min at 55 °C. The eluted DNA was subsequently purified via phenol:chloroform:isoamylalcohol (pH 8.0, Sigma P2069) and then ethanol precipitated. For downstream applications, the DNA was either analysed by qPCR using Fast SYBR Green (Thermo, 4385618) or used for DRIP-seq.

### DRIP-seq
5 ng of DRIP DNA from each of three biological replicates from each condition were used for library preparation using the Thruplex DNA-seq kit for Illumina (TAKARA, R400674) according to the manufacturer's protocol using 12 PCR cycles for IP samples, and 8 PCR cycles for input samples. Libraries were then sequenced as 2 lanes of S1 PE50 on the Illumina NovaSeq6000.

### DRIP-Seq and CUT&Tag analysis
Raw fastq files were filtered to select high quality reads by using fastp[70] and aligned to the hg38 human reference genome with Bowtie2[71]. For peak calling of CnT alignments, MACS2[72] was employed using WT cells as the test condition and KLF5 KO cells as

the control condition, identifying KLF5-dependent peaks. To improve peak calling accuracy, peaks were first filtered for those in 2 out of 3 technical replicates per biological replicate, and then in 2 of 3 biological replicates. Peaks were further filtered to only maintain those with a signal value greater than 15. Bigwig coverage files were generated using Deeptools[73] for the creation of heatmaps, coverage and metagene plots. To analyse coverage per gene, bedtools[74] was used to convert alignment files to bed files and to calculate coverage per gene which was then normalised to the number of aligned reads. Per-gene coverage data was loaded into R for downstream statistical analysis and plot generation.

### Reporting summary
Further information on research design is available in the Nature Portfolio Reporting Summary linked to this article.

## Data availability
All sequencing data generated in this study including NormZ scores arising from DrugZ analysis of the CRISPR-Cas9 screens, differential expression and gene set enrichment analysis arising from RNA-sequencing are provided in Supplementary Data 1 and Supplementary Data 2. Sequencing reads for the CRISPR screen have been deposited in European Nucleotide Archive (ENA) under accession number PRJEB75512. RNA sequencing data generated in this study have been deposited in ENA under accession number PRJEB75523 [https://www.ebi.ac.uk/ena/browser/view/PRJEB75523]. DRIP-seq raw data generated in this study have been deposited with Array Express under the accession code E-MTAB-14062. Cut and Tag raw data generated in this study have been deposited with Array Express under the accession code E-MTAB-14079. All other data are available from the corresponding authors upon request. Newly generated biological materials, including cell lines described in this study are available from the corresponding authors, subject to a suitable Material Transfer Agreement (MTA) where relevant. Source data are provided with this paper.

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

## Acknowledgements

We thank C.J. Carnie for critical reading of the manuscript, K. Dry for editorial assistance and all members of the S.P.J. laboratory for advice and discussions. We also acknowledge K. Harnish of the Gurdon Institute in-house sequencing facility for performing next-generation sequencing of our CRISPR-Cas9 screening samples and all members of the Gurdon and Cancer Research UK Cambridge Institute core facilities for assistance and support. Research in the S.P.J. laboratory is supported by Cancer Research UK (CRUK) Discovery Award DRCPGM\100005, CRUK core grant C9545/A29580 and ERC Synergy Award 855741 (DDREAMM). S.W.A. is supported by a Mark Foundation for Cancer Research (MFCR) ASPIRE II Award, is a recipient of the Women's Postdoctoral Career Development Award in Science from the Weizmann Institute of Science and was a recipient of an Outstanding Postdoctoral Women Fellowship from the Israeli Council for Higher Education. The lab was also supported by CRUK Programme grant C6/A18796 and core funding grants C6946/A24843 and WT203144 to the Gurdon Institute. A.S.B. was supported by CRUK RadNET Cambridge C17918/A28870 and Wellcome Early Career Award 227014/Z/23/Z, N.G by MFCR ASPIRE I, V.G. by Wellcome Investigator Award, 206388/Z/17/Z, S.L. and T.A.T by ERC Synergy Award 855741, J.C. by CRUK core funding and MFCR ASPIRE II, and R.B. by CRUK Discovery Award DRCPGM\100005. S.P.J. receives a salary from the University of Cambridge. Work in the S.B. laboratory is supported by CRUK (C9545/A19863) and programme award funding (C9681/A29214) and Herchel Smith Funds. S.B. is a Senior Investigator of the Wellcome Trust (209441/Z/17/Z).

## Author contributions

S.W.A. conceived the study, planned all the experiments, analysed data, and wrote the original draft. C.D. performed Cut & Tag experiments. N.G. and J.C. generated cell lines and performed experiments. A.S.B., S.L., V.G. and T.A.T. performed bioinformatic analyses, and QIBC. R.B. helped in editing the manuscript. S.B. supervised work by C.D. S.W.A. and S.P.J. coordinated and supervised the project and co-wrote the manuscript.

## Competing interests

S.B. is a founder and shareholder of Biomodal Ltd., Inflex Ltd. and RNAvate Ltd. S.P.J works part time as Chief Research Officer at Insmed Innovation UK Ltd. S.P.J is a founding partner of Ahren Innovation Capital LLP and is co-founder, a board member, and Chair of the Scientific Advisory Board of Mission Therapeutics Ltd. S.P.J. is a consultant and shareholder of Inflex Ltd. The authors declare no other competing interests.
