## [Transparent Peer Review file · Nature Communications]

KLF5 loss sensitizes cells to ATR inhibition and is synthetic lethal with ARID1A deficiency

Corresponding Author: Professor Stephen Jackson

Version 0:

Reviewer comments:

Reviewer #1

(Remarks to the Author)

In the manuscript titled "KLF5 loss sensitizes cells to ATR inhibition and is synthetic lethal with ARID1A deficiency", Samah et al. conducted genome-scale CRISPR screening to reveal that transcription factor KLF5 loss sensitizes ARID1A WT, but not ARID1A KO cells to ATR inhibition. In contrast, KLF5 is required for the proliferation and colony forming ability of ARID1A KO cells. The authors successfully demonstrate that KLF5 deletion increases replication stress, DNA damage and R-loop formation in ATRi-treated ARID1A WT cells. The authors show that KLF5 loss results in failure to recruit BRD4 to chromatin and that depletion of BRD4 sensitizes KLF5 WT, but not KLF5 KO cells to ATR inhibition. This is consistent with previous reports establishing a role for BRD4 in R-loop homeostasis through RNA Polymerase II elongation. Overall, the findings in this manuscript establish a role for KLF5 in ATRi-dependent cell stasis. However, the data assertion that KLF5 is a synthetic lethal target in ARID1A mutant cells is underdeveloped and the molecular mechanism is completely absent. I would suggest removing data in connection with ARID1A mutation or strengthening these claims. In addition, the title would need to be changed to reflect the actual emphasis of the paper.

Specific concerns are listed below:

Major points:

1. Is KLF5 universally expressed? Since KLF5 is one of a large family of KLF transcription factors with cell type specific expression, it's unclear whether KLF5 would be universally required for BRD4 recruitment, or simply in this cell line. The authors should discuss the role of other KLF family members in this function.
2. Figure 4C-D, representative images should be provided. It is also unclear why two different formats of plots were used between C and D even though they are similar experiments.
3. To be relevant in translational settings, it would be important to know what cell types/cell lines might be sensitized to ATRi by KLF5 depletion. What types of cancer cell lines are dependent on KLF5? Relevant papers from DepMap related to KLF5 dependency should be cited in discussion paragraph 2. Are KLF5/ATR inhibitors additive/synergistic?
4. Throughout the paper, U2-OS cells are used, with one panel showing RPE-1 proliferation curves. Other cell lines should be used, particularly to investigate point 3, i.e. in what cancer types KLF5 depletion/inhibition could function to sensitize ATRi resistant cancers.
5. Was BRD4 found in the authors' original screen?

For ARID1A mutant synthetic lethality:

1. In addition to colony forming abilities, cell proliferation should also be tested using a proliferation assay.
2. Is KLF5 important for recruiting BRD4 and/or ARID1B to chromatin in ARID1A mutant cells?
3. Are ARID1A mutant human cell lines sensitive to KLF5 deletion in DepMap studies?
4. Are KLF5 inhibitors effective at killing/slowing the growth of ARID1A mutant versus ARID1A WT cells?
5. Do ARID1A/KLF5 mutant cells die from increased replication stress, DNA damage, apoptosis?

Minor points:

1. KLF5 is misspelled in many instances.

Reviewer #2

(Remarks to the Author)

Reviewer #3

(Remarks to the Author)

In this study, the authors performed CRISPR-Cas9 screens to identify genes whose mutations differentially influence sensitivity to ATR inhibitors (ATRi) in wild-type (WT) versus ARID1A knockout (KO) cells. They then characterized one hit in more detail, KLF5, whose inactivation appears to sensitize WT (but much less ARID1A) cells to ATRi. Overall, the study is well-executed and the results are clearly presented; however, certain conclusions need stronger support and clearer explanations. The paper introduces two main concepts: first, that ATRi increases R-loops and transcription-replication conflicts (TRC) in KLF5 KO cells, and second, that there exists a synthetic lethality between KLF5 and ARID1A KO. The latter concept should be better integrated into the paper's narrative, as detailed below. Additionally, I note that the study frequently connects known facts, repackaging them within the context of the above-mentioned CRISPR screens. Because of this, some might argue that certain results were somewhat anticipated based on existing literature. Whether this level of novelty is sufficient for publication in Nature Communications is an editorial decision.

For example, ARID1A is known to modulate R-loops (PMID: 33826602), as is BRD4 (PMID: 32796829). The link between BRD4 activity and KLF5 has also been previously published (PMID: 33115806). Consistent with the above, the authors demonstrate that the absence of KLF5 elevates R-loops; given KLF5's impact on BRD4 function, one might argue that this result was a bit expected. Furthermore, the authors state: "These observations support previous findings showing that cells lacking ARID1A are dependent on BRD4-driven transcription for growth, and that BRD4 inactivation by using JQ1 inhibitors is toxic to ARID1A-mutant cells." Considering this and the fact that KLF5 is known to promote BRD4 function, it was also anticipated that the absence of KLF5 would be detrimental to ARID1A KO cells. Nevertheless, I acknowledge that providing experimental proof that this indeed occurs is certainly not devoid of interest.

As previously mentioned, the title of the manuscript highlights the synthetic lethality between ARID1A and KLF5 loss; however, this is a comparatively small part of the current manuscript. Presenting additional characterization of the mechanistic basis of this synthetic lethality (potentially due to elevation in R-loop dysregulation?) would be beneficial. Indeed, it feels plausible that combining two knockouts in genes affecting R-loop regulation (KLF5 and ARID1A) may exacerbate this phenotype, thereby compromising growth. In my opinion, the exact mechanism underlying the synthetic lethality was not explored in sufficient detail; there is also no clear explanation of this synthetic lethality in the model presented in Figure 6F. Moreover, in the discussion, the authors state, "Of note, data from the DepMap project indicate that various cancer cell lines are particularly dependent on KLF5 expression, thereby highlighting how targeting KLF5 in combination with ATRi could provide an opportunity for treating KLF5-dependent cancer cells." In my opinion, the authors should use DepMap to evaluate whether cancer cell lines lacking ARID1A are generally sensitized to KLF5 knockout, which would provide further support for their results.

Importantly, the authors do not explicitly state why the lack of ARID1A would limit the impact of KLF5 knockout on sensitivity to ATR inhibitors (ATRi) (Figure 1D). Since ARID1A knockout cells already display elevated R-loops, is it possible that the impact of KLF5 is reduced? Is this the proposed model? Similarly, it is not clarified how the authors rationalize the notion that ATRi or Chk1 inhibitors (Chk1i) elevate R-loop formation/TRCs specifically in cells lacking KLF5 (Fig. 4A and D). Curiously, based on Fig. 4A, transcription-replication collisions (TRCs) appear to diminish upon ATRi treatment in WT cells. The mechanism linking ATRi/Chk1i to TRC/R-loop formation, both in general and in cells lacking KLF5, should be discussed in more detail. For example, certain papers already report functional links between ATR and R-loops suppression (e.g., PMID: 37930853, PMID: 28314779).

The authors present an RNA-seq analysis of the transcriptome in cells lacking KLF5 versus WT. Beyond a somewhat limited pathway analysis, this data is not used further and appears disconnected from the rest of the study. In my opinion, it would be important to evaluate whether transcriptional dysregulation caused by KLF5 at specific genes is associated with abnormal formation of R-loops specifically at those genes. Additionally, is the increase in R-loops caused by KLF5 knockout strictly observed at genes where BRD4 recruitment is dysregulated? Further analysis of the data in this regard is warranted: this could be achieved by a more thorough comparison between the DRIP-seq/RNA-seq data and the KLF5/BRD4 cut-and-tag experiments. As it stands now, the authors imply that dysregulated transcription in KLF5 knockout cells is responsible for elevated R-loop formation, but a direct link between these points is lacking. While inhibition of transcription is shown by the authors to reduce R-loops in KLF5 knockout cells, this would be expected regardless of whether KLF5 directly modulates R-loops at specific genes, or acts through other more indirect mechanisms. Finally, it would also be of interest to check and comment on whether the expression of known R-loop regulators is modulated in KLF5 knockout cells versus WT at the transcriptional level (using RNA-seq data).

Specific points related for Figures:

-In Figure 3, it would be of interest to evaluate the incorporation of EdU (the levels/intensity of EdU incorporation and not

only the fraction of cells that are EdU+) specifically in γ -H2AX high cells; one would expect that if lesions are occurring specifically in S phase, EdU incorporation might be reduced. The authors should also comment on why cells would accumulate in G1, and not S or G2/M, if DNA lesions are accumulating during S phase in KLF5 knockout cells treated with ATRi. Additionally, the authors state, "KLF5 depleted cells accumulate more DNA damage and therefore withdraw at a higher rate from S-phase." It is not clear what is meant by "withdrawing" from S phase and why damage in S would facilitate S phase progression/exit; a better discussion and explanation of these data and their significance is warranted. Finally, I note that the increase in DNA damage is limited in these cells (7-18% of cells are γ -H2AX high); as a control, it would be interesting to compare this with BRD4 siRNA or other conditions known to limit R-loops. Is this level of γ -H2AX accumulation expected for a bona fide R-loop regulator?

-In figure 3, the authors report elevated γ -H2AX after 24h of ATRi treatment. I wonder if this could be due, at least in part, to cell death/apoptosis which can cause caspase-dependent elevation in γ -H2AX, which would change the interpretation of the data. Evaluating apoptosis (annexin V staining for example) and testing caspase-dependency of γ -H2AX signal would clarify this.

-In Figure 4A, a control in WT cells with ATRi and DRB is missing from these experiments. It is also curious to see that ATRi reduces TRCs in WT cells. A statistical analysis of this should be presented and commented on if significant. This relates to a previous point regarding the need for the authors to at least briefly comment on why TRCs are elevated in KLF5 knockout cells treated with ATRi or Chk1i, but reduced in WT cells. This seems contradictory with published data indicating that ATR prevents R-loop generation via phosphorylation of key regulators, e.g., PMID: 28314779, PMID: 37930853.

-In Figure 4, the authors show elevated R-loops in cells lacking KLF5. It is worth considering whether the increase in TRCs and R-loops might be linked to increased cell death. Is this a specific consequence of KLF5 loss, or is it due to lethality? One possible way to evaluate this would be to use an equitoxic dose of ATRi in WT cells to assess the impact on R-loop formation.

-In figure 4: the increase in TRCs in KLF5 KO treated with ATRi vs DMSO is not supported by a statistical test (Fig. 4A).

-Figure 5A and B appear redundant; one of the panel could be in supplemental.

-The input (total lysate) should be presented in Fig. 6A and S5A.

-In figure 6E: a statistical test between KLF5 knockout (KO) with and without siBRD4 would be appropriate to determine if the difference is significant. The current grouping of samples for statistical purposes is not very informative regarding the point being made.

Minor points:

-Page 2: "This also increases conflicts between the replisome and the transcription machinery, associated with formation of DNA-RNA hybrids (R-loops; 11, leading to genome instability and threatening cell survival through replication-fork collapse 12,13."; the parenthesis needs to be closed.

-The last section's header is "KLF5 loss impairs BRD4 chromatin recruitment and is lethal for ARID1A-null cells", however, no explicit discussion of the synthetic lethality is made in this section.

Reviewer #4

(Remarks to the Author)

The manuscript by Awwad et al describes the discovery of KLF5 as a synthetic lethal interactor of ARID1A, and further shows that loss of KLF5 sensitizes cells to ATR inhibitors. Mechanistically, the authors show that treatment of KLF5-depleted cells with ATR inhibitors causes replication stress, transcription-replication collisions, and defective BRD4 localization. Overall, the manuscript will be a welcomed addition to the field, with important findings and convincing results.

Specific comments

1. While the manuscript is set up by a CRISPR screen identifying ARID1A as synthetic lethal with KLF5, after a brief validation of this genetic interaction, the authors focus on investigating KLF5 in the response to ATR inhibitors. There is somewhat of a disconnect here. I would also note that some aspects of the manuscript's novelty are impacted by previously-published findings that KLF5 promotes BRD4 localization, and that BRD4 is synthetic lethal with ARID1A. Thus, any additional mechanistic insights that can be provided would increase the manuscript's novelty. The authors could investigate if the synthetic lethality between ARID1A and KLF5 (as well as BRD4) correlates with replication-transcription conflicts, increased R-loops, increased γ H2AX/RPA foci, and is suppressed by RNaseH1 overexpression.

2. The validation of the screen results in performed only in U2OS cells. Additional results need to be presented in the support of the statement in the last sentence of the Abstract (KLF5 is a target for ARID1A-deficient cancers). The authors should be able to easily confirm the synthetic lethality in other cell lines (they already have KLF5-knockout RPE1 cells). They can also use the KLF5 inhibitor (ML264) to investigate if this causes increased cell death in ARID1A-deficient cells.

3. In my view, a significant weakness of the paper is that the majority of the mechanistic work (Fig 3-6) was performed in a single cell line, using a single KLF5 knockout clone (U2OS KO5). In addition, a subset of the presented experiments are inconsistently performed with siRNA (gH2AX after ATRi in 3C is done with KO while CHK1i in 3D is done with siRNA). Some of the experiments in these figures need to be shown with multiple KO clones (or both KO and siRNA) and in additional cell lines, to enhance the rigor of the study.

4. Confirming the results in additional cell lines is also important in light of the fact the observed impact of KLF5-KO, albeit statistically significant, appears minimal. For example, through multiple figures, the % cells with gH2Ax staining and the % cells with RC go up from 1% to 5%. Is this increase sufficient to explain the loss of viability? It seems somewhat unlikely. Does increasing the ATRi concentration or the treatment time result in a more drastic increase/difference?

5. There are no statistical analyses shown in Fig 3A.

Reviewer #5

(Remarks to the Author)

Version 1:

Reviewer comments:

Reviewer #1

(Remarks to the Author)

The authors have addressed my concerns. However, I would advise adding two references PMID: 33986538, 38754421 to the discussion in lines 421 "supported by ARID1A's function in modulating R-loops 54".

Reviewer #2

(Remarks to the Author)

Reviewer #3

(Remarks to the Author)

The authors have answered my concerns satisfactorily, and in my opinion the current version is suitable for publication.

Reviewer #4

(Remarks to the Author)

The revised manuscript addresses my comments and, in my opinion, the other reviewers' comments as well. The authors added a large number of additional experiments that validate their initial findings and reveal new mechanistic insights underlying the observed synthetic lethality.

Response to the reviewers

We very much appreciate the time and efforts that the reviewers and the editor have taken in evaluating our manuscript, and we thank them for their thorough assessments of our work. Indeed, the reviewers' comments and suggestions prompted us to carry out experiments that we might not otherwise have done. As described in detail below, we have been able to address all the points raised by the reviewers, leading to a revised and improved manuscript, now submitted, which contains seven main figures and seven supplementary figures.

This document includes:

- (1) Summary of the main new findings of the revised manuscript.
- (2) Point-to-point responses to the reviewers' comments.

1) **Summary of the main new data presented in the revised manuscript:**

- We have provided new data showing that KLF5 depletion/inhibition sensitizes *ARID1A*-deficient cells:
 - (i) We show that siRNA-depletion of KLF5 reduces cell viability in *ARID1A* KO RPE-1 compared to WT cells (**new Supplementary Fig. 1d**).
 - (ii) We show that KLF5 inhibition significantly reduces the proliferation of U2-OS, MCF7, and CAL-51 *ARID1A*-deficient versus WT cells (**new Supplementary Fig. 1e-h**).
 - (iii) Using DepMap data, we show that *ARID1A* mutant cancer cell lines are particularly sensitive to losing KLF5, as evidenced by the negative CRISPR score (**new Supplementary Fig. 1i**).
- We show that, in addition to U2-OS and RPE-1 cells, KLF5 depletion hypersensitizes HAP-1 cells to ATRi (**new Fig. 2d and Supplementary Fig. 2e**).
- We have confirmed the phenotypes of KLF5 loss on cell cycle profile, DNA damage and replication catastrophe upon ATRi in both U2-OS and RPE-1 cells (**new Fig. 5c, Supplementary Fig. 3a-g, and Supplementary Fig. 5b-c**).
- We provide new data supporting the molecular mechanism of the synthetic lethality between *ARID1A* and KLF5, showing that KLF5 and *ARID1A* co-depletion increases R-loop-dependent DNA damage (**new Supplementary Fig.7**).

2) **Point-to-point responses to the reviewers' comments.**

Our responses are in the order that they were raised in the reviewers' reports, with reviewers' comments in black and **our responses in green**.

Reviewer #1 (Remarks to the Author):

In the manuscript titled "KLF5 loss sensitizes cells to ATR inhibition and is synthetic

lethal with ARID1A deficiency”, Samah et al. conducted genome-scale CRISPR screening to reveal that transcription factor KLF5 loss sensitizes ARID1A WT, but not ARID1A KO cells to ATR inhibition. In contrast, KLF5 is required for the proliferation and colony forming ability of ARID1A KO cells. The authors successfully demonstrate that KLF5 deletion increases replication stress, DNA damage and R-loop formation in ATRi-treated ARID1A WT cells. The authors show that KLF5 loss results in failure to recruit BRD4 to chromatin and that depletion of BRD4 sensitizes KLF5 WT, but not KLF5 KO cells to ATR inhibition. This is consistent with previous reports establishing a role for BRD4 in R-loop homeostasis through RNA Polymerase II elongation. Overall, the findings in this manuscript establish a role for KLF5 in ATRi-dependent cell stasis. However, the data assertion that KLF5 is a synthetic lethal target in ARID1A mutant cells is underdeveloped and the molecular mechanism is completely absent. I would suggest removing data in connection with ARID1A mutation or strengthening these claims. In addition, the title would need to be changed to reflect the actual emphasis of the paper.

We thank this reviewer for evaluating our work, and for raising these important issues. We now include more data supporting our findings related to the synthetic lethality between ARID1A and KLF5 (please see below).

Specific concerns are listed below:

Major points:

1. Is KLF5 universally expressed? Since KLF5 is one of a large family of KLF transcription factors with cell type specific expression, it's unclear whether KLF5 would be universally required for BRD4 recruitment, or simply in this cell line. The authors should discuss the role of other KLF family members in this function.

We thank the reviewer for raising these important points and agree that it is important to explore them further.

Data from the Human Protein Atlas show that KLF5 is universally expressed across human tissues (https://www.proteinatlas.org/ENSG00000102554-KLF5/tissue#rna_expression), and it is overexpressed in various types of human cancer. Moreover, we and others have shown that KLF5 promotes the chromatin recruitment of BRD4 in multiple cell backgrounds. Our data provide evidence of such function in the osteosarcoma cell line, U2-OS, and in the untransformed human retinal pigment epithelial-1 (RPE-1) cell line, and previous work has shown this in the lung squamous cancer cell line, HARA (PMID 33115806). These observations therefore suggest that KLF5 is widely, if not universally, required for recruiting BRD4, rather than cell-line/type specific.

KLF5 indeed belongs to the KLF transcription factor family, which in humans consists of 17 members. Despite the similarity in their overall protein structure, whether other KLFs are involved in regulating BRD4 recruitment to chromatin has not been experimentally demonstrated. However, our CRISPR screen data suggest that regulation of BRD4 and impacts on cellular responses to ATRi are largely specific to KLF5 – that is, if other KLFs are redundantly involved in such functions, this would have obscured the impacts of KLF5 deficiency that we have clearly observed. We address these issues in our revised text (**new Supplementary Fig. 1i, and page 15**).

2. Figure 4C-D, representative images should be provided. It is also unclear why two different formats of plots were used between C and D even though they are similar experiments.

In our revised manuscript, we now include representative images, and we present them using the same format. These figures are now moved to supplementary information (**new Supplementary Fig. 4a-c**) due to overlap with the DRIP-seq data in Figure 4.

3. To be relevant in translational settings, it would be important to know what cell types/cell lines might be sensitized to ATRi by KLF5 depletion. What types of cancer cell lines are dependent on KLF5? Relevant papers from DepMap related to KLF5 dependency should be cited in discussion paragraph 2. Are KLF5/ATR inhibitors additive/synergistic?

We appreciate the reviewer's interest in the potential importance of our findings in translational and clinical settings. However, there is not enough data concerning ATR inhibition in DepMap studies nor in our recent CRISPR screens web portal (<https://sjlab.cruk.cam.ac.uk/ddrcs/home>) to reasonably model the effect of KLF5 loss in response to ATRi in different cell types and to determine whether KLF5/ATR inhibitors are additive/synergistic. We hope that this becomes possible in future.

On the other hand, we took advantage of DepMap data and assessed KLF5-dependency across cancer types (see figure below) and cited the relevant papers in the discussion of our revised manuscript (**page 15**). Such analysis shows that several cancer types are dependent on KLF5 expression, which suggests that a combination of KLF5 depletion/inhibition and ATR inhibitors could be a promising strategy for treating such cancer cells. However, whether KLF5 depletion and ATR inhibition is synergistic, or additive requires more data, which are currently unavailable.

The figure above shows the distribution of KLF5-dependent cancer cell lines. Gene dependency data were curated from the DepMap portal to analyse the compositions

of cancer cell lines depending on KLF5 expression. Cell lines with dependency probability of ≥ 0.5 were selected, resulting in 347 cell lines being dependent on KLF5.

4. Throughout the paper, U2-OS cells are used, with one panel showing RPE-1 proliferation curves. Other cell lines should be used, particularly to investigate point 3, i.e. in what cancer types KLF5 depletion/inhibition could function to sensitize ATRi resistant cancers.

In our original manuscript, we validated the CRISPR screen results showing that loss of KLF5 sensitizes cells to ATRi in U2-OS and RPE-1 cells. In response to the reviewer's request, we now add new data showing that loss of KLF5 hypersensitizes HAP-1 cells (derived from KBM-7 chronic myelogenous leukaemia cell line) to ATRi (**new Fig. 2d and Supplementary Fig. 2e**).

To answer what cancer types KLF5 depletion could function in to sensitize ATRi resistant cancers, our results show that loss of KLF5 is associated with increased transcription-replication conflicts (TRCs) and R-loops upon ATRi treatment. Therefore, we anticipate that if ATRi-resistance arises in certain genetic backgrounds due to a reduction in TRCs and R-loops, then loss of KLF5 may resensitize these ATRi-resistant backgrounds. Indeed, previous work from our lab (PMID: 34329458) showed that loss of CCNC or CDK8, subunits of the RNA polymerase II mediator complex reduces TRCs and R-loops and provide ATRi resistance. Prompted by this, we tested the effect of CRISPR sgRNA-mediated KLF5 depletion in HAP-1-*CDK8* KO cells. The results showed that KLF5 depletion re-sensitized HAP-1-*CDK8* KO cells to ATR inhibition. These results thus support the notion that increased TRCs and R-loops is a vulnerability towards ATRi, and provides evidence that targeting KLF5 might be used to circumvent ATRi resistance. These results are part of an ongoing project in our lab related to understanding the mechanisms for sensitizing ATRi-resistant backgrounds, and therefore we are not including them in the current manuscript.

The figure above shows results of alamarBlue cell proliferation assays in WT, *CDK8* KO, or *CDK8* KO HAP-1 cells transduced with sgRNAs (#2 and #3) targeting KLF5 and treated with ATRi. Data are represented as survival curves on the left and AUCs on the right. Error bars = mean \pm SEM (biological n = 3).

5. Was BRD4 found in the authors' original screen?

We thank the reviewer for bringing up this question. BRD4 does not drop-out in our U2-OS CRISPR screen combined with ATRi, or in other CRISPR screens using the Brunello sgRNA library (<https://sjlab.cruk.cam.ac.uk/ddrcs/home>). This might be due to various reasons, among them the efficiency and the activity of the sgRNAs targeting BRD4 in the Brunello library. On the other hand, BRD4 drops out with better FDRs using other libraries such as TKOv2 and TKOv3 (<https://sjlab.cruk.cam.ac.uk/ddrcs/home>). Having said this, we noticed that although mutations in ARID1A are markers for ATR inhibitor sensitivity, ARID1A also does not drop out in CRISPR screens combined with ATRi (<https://sjlab.cruk.cam.ac.uk/ddrcs/home>).

For ARID1A mutant synthetic lethality:

1. In addition to colony forming abilities, cell proliferation should also be tested using a proliferation assay.

We have generated RPE-1 *ARID1A* KO cells and tested the effect of siRNA-mediated KLF5 depletion on cell proliferation. In our revised manuscript, we show that depletion of KLF5 reduces cell viability of *ARID1A* KO compared to WT cells (**new Supplementary Fig. 1d**). In addition, we have assessed the effect of KLF5 inhibition on the cell proliferation of several ARID1A mutant cell lines (please see our detailed response for point 4).

2. Is KLF5 important for recruiting BRD4 and/or ARID1B to chromatin in ARID1A mutant cells?

This is a very good point. We performed chromatin fractionation in WT, *ARID1A* KO, and *ARID1A* KO RPE-1 cells transfected with KLF5 siRNA and tested for effects on BRD4 chromatin-bound fraction. As shown in our revised manuscript, we found that, indeed, KLF5 promotes the recruitment of BRD4 to chromatin in *ARID1A* KO cells (**new Supplementary Fig. 7a**). These results further support our results relating to the role of BRD4 in driving the novel synthetic lethality between ARID1A and KLF5.

3. Are ARID1A mutant human cell lines sensitive to KLF5 deletion in DepMap studies?

Again, we thank the reviewer for raising this important point. Using DepMap data, we examined the effect of *KLF5* gene knockout on *ARID1A*-deficient cancer cell lines. As shown in **new Supplementary Figure 1i** of our revised manuscript, targeting *KLF5* in *ARID1A*-deficient cancer cells has a negative CRISPR score, which suggests that ARID1A mutant cells are sensitive to KLF5 depletion. As a positive control, we examined the effect of targeting ARID1B, a known synthetic lethal partner of ARID1A. Moreover, we assessed the effect of other transcription factors: KLF4 as another member of the KLF family, and E2F7 as a more broadly used transcription factor. Targeting either KLF4 or E2F7 does not appear to influence *ARID1A* mutant cell lines. These analyses further strengthen our observations and conclusions that KLF5 is a synthetic lethal target for ARID1A-deficient cells and implies the specificity of KLF5 in such settings.

4. Are KLF5 inhibitors effective at killing/slowing the growth of ARID1A mutant versus ARID1A WT cells?

In response to this question, we assessed the effect of KLF5 inhibition on the proliferation of U2-OS and MCF-7 *ARID1A* WT and KO cells. As shown in **new Supplementary Figure 1e-f**, KLF5 inhibition showed a significant reduction in the growth of *ARID1A* KO versus *ARID1A* WT cells.

Moreover, we took advantage of the triple negative breast cancer cell line, CAL-51, that does not express ARID1A and complemented these cells with an empty vector or a vector expressing WT ARID1A (**new Supplementary Fig. 1g**). As shown in **new Supplementary Figure 1h**, complementation with WT ARID1A partially rescued the slow growth observed upon KLF5 inhibition in CAL-51 cells. It should be noted that the observed reduction in the growth of ARID1A mutant cells treated with KLF5 inhibitor is not large, yet is significant, and might be related to the mode of action, specificity and other factors affecting the efficiency of the KLF5 inhibitor (ML264) used here.

5. Do ARID1A/KLF5 mutant cells die from increased replication stress, DNA damage, apoptosis?

We thank the reviewer for raising this important question. We provide several lines of evidence showing that ARID1A and KLF5 die from increased R-loop-dependent DNA damage.

Firstly, we performed RNA-seq in U2-OS *ARID1A* KO cells and performed a comprehensive analysis between the transcriptional profiles of *ARID1A* KO and *KLF5* KO cells, focusing on their effects on regulating the expression of *bona fide* regulators of R-loops. Such analysis revealed that both proteins are involved in regulating the expression of several regulators of R-loops such as DHX9, BRCA1, FANCD2 and TREX1 (**new Supplementary Fig. 7b**). Furthermore, we co-depleted ARID1A and KLF5 and assessed the DNA damage marker γ H2AX in the absence or presence of overexpressing WT RNaseH1. As shown in **new Supplementary Figure 7c**, co-depletion of ARID1A and KLF5 led to increased γ H2AX (2.4-fold increase) signal when compared to control, more than the inductions seen upon the single depletion of either protein, and this increase in γ H2AX was diminished upon WT RNaseH1 overexpression (2.1 fold-decrease). These results therefore suggest that both ARID1A and KLF5 are involved in the homeostasis of R-loops, and their combined depletion perturbs R-loop regulation sufficiently to cause accumulation of DNA damage and ensuing cell death/slow cell growth. Adding to this, we provide evidence that KLF5 is required for recruiting BRD4 to chromatin in ARID1A KO cells (**new Supplementary Fig. 7a**). Collectively, and consistent with BRD4's role in preventing the accumulation of R-loops, these results suggest that ARID1A and KLF5 mutant cells die from increased R-loop-dependent DNA damage.

Minor points:

1. KLF5 is misspelled in many instances.

We apologise and thank the reviewer for spotting this. We have now corrected these typos in the revised manuscript.

Reviewer #2 (Remarks to the Author)

We thank reviewer 2 for taking part in the review process for our paper.

Reviewer #3 (Remarks to the Author)

In this study, the authors performed CRISPR-Cas9 screens to identify genes whose mutations differentially influence sensitivity to ATR inhibitors (ATRi) in wild-type (WT) versus ARID1A knockout (KO) cells. They then characterized one hit in more detail, KLF5, whose inactivation appears to sensitize WT (but much less ARID1A) cells to ATRi. Overall, the study is well-executed and the results are clearly presented; however, certain conclusions need stronger support and clearer explanations. The paper introduces two main concepts: first, that ATRi increases R-loops and transcription-replication conflicts (TRC) in KLF5 KO cells, and second, that there exists a synthetic lethality between KLF5 and ARID1A KO. The latter concept should be better integrated into the paper's narrative, as detailed below. Additionally, I note that the study frequently connects known facts, repackaging them within the context of the above-mentioned CRISPR screens. Because of this, some might argue that certain results were somewhat anticipated based on existing literature. Whether this level of novelty is sufficient for publication in Nature Communications is an editorial decision.

For example, ARID1A is known to modulate R-loops (PMID: 33826602), as is BRD4 (PMID: 32796829). The link between BRD4 activity and KLF5 has also been previously published (PMID: 33115806). Consistent with the above, the authors demonstrate that the absence of KLF5 elevates R-loops; given KLF5's impact on BRD4 function, one might argue that this result was a bit expected. Furthermore, the authors state: "These observations support previous findings showing that cells lacking ARID1A are dependent on BRD4-driven transcription for growth, and that BRD4 inactivation by using JQ1 inhibitors is toxic to ARID1A-mutant cells." Considering this and the fact that KLF5 is known to promote BRD4 function, it was also anticipated that the absence of KLF5 would be detrimental to ARID1A KO cells. Nevertheless, I acknowledge that providing experimental proof that this indeed occurs is certainly not devoid of interest.

As previously mentioned, the title of the manuscript highlights the synthetic lethality between ARID1A and KLF5 loss; however, this is a comparatively small part of the current manuscript. Presenting additional characterization of the mechanistic basis of this synthetic lethality (potentially due to elevation in R-loop dysregulation?) would be beneficial.

Indeed, it feels plausible that combining two knockouts in genes affecting R-loop regulation (KLF5 and ARID1A) may exacerbate this phenotype, thereby compromising growth. In my opinion, the exact mechanism underlying the synthetic

lethality was not explored in sufficient detail; there is also no clear explanation of this synthetic lethality in the model presented in Figure 6F.

We thank the reviewer for the in-depth evaluation of our manuscript, and for bringing up various key issues and questions.

Related to the mechanistic basis of the synthetic lethality between ARID1A and KLF5, we have included this part of our manuscript in the model presented in **new Figure 7**, which has been further supported by experimental data. Firstly, we performed RNA-seq in U2-OS *ARID1A* KO cells and performed a comprehensive analysis between the transcriptional profiles of *ARID1A* KO and *KLF5* KO cells, focusing on their effects on regulating the expression of *bona fide* regulators of R-loops. Such analysis revealed that both proteins are involved in regulating the expression of several regulators of R-loops such as DHX9, BRCA1, FANCD2 and TREX1 (**new Supplementary Fig. 7b**). Furthermore, we co-depleted ARID1A and KLF5 and assessed the DNA damage marker γ H2AX in the absence or presence of overexpressing WT RNaseH1. As shown in **new Supplementary Figure 7c**, co-depletion of ARID1A and KLF5 led to increased γ H2AX (2.4-fold increase) signal when compared to control, more than the inductions seen upon the single depletion of either protein, and this increase in γ H2AX was diminished upon WT RNaseH1 overexpression (2.1 fold-decrease). These results therefore suggest that both ARID1A and KLF5 are involved in the homeostasis of R-loops, and their combined depletion perturbs R-loop regulation sufficiently to cause accumulation of DNA damage and ensuing cell death/slow cell growth. Adding to this, we provide evidence that KLF5 is required for recruiting BRD4 to chromatin in ARID1A KO cells (**new Supplementary Fig. 7a**). Collectively, and consistent with BRD4's role in preventing the accumulation of R-loops, these results suggest that ARID1A and KLF5 mutant cells die from increased R-loop-dependent DNA damage.

Moreover, in the discussion, the authors state, "Of note, data from the DepMap project indicate that various cancer cell lines are particularly dependent on KLF5 expression, thereby highlighting how targeting KLF5 in combination with ATRi could provide an opportunity for treating KLF5-dependent cancer cells." In my opinion, the authors should use DepMap to evaluate whether cancer cell lines lacking ARID1A are generally sensitized to KLF5 knockout, which would provide further support for their results.

We agree; this is a good suggestion. In our revised manuscript, we now include data from DepMap studies showing that, indeed, ARID1A mutant cancer cells are particularly sensitive to losing KLF5. As shown in **new Supplementary Figure 1i** of our revised manuscript, targeting *KLF5* in *ARID1A*-deficient cancer cells has a negative CRISPR score. As a positive control, we examined the effect of targeting ARID1B, a known synthetic lethal partner of ARID1A. Moreover, we assessed the effect of other transcription factors: KLF4 as another member of the KLF family, and E2F7 as a more broadly used transcription factor. Targeting either KLF4 or E2F7 does not appear to influence *ARID1A* mutant cell lines. These analyses further strengthen our observations and conclusions that KLF5 is a synthetic lethal target for ARID1A-deficient cells and implies the specificity of KLF5 in such settings.

Importantly, the authors do not explicitly state why the lack of ARID1A would limit the impact of KLF5 knockout on sensitivity to ATR inhibitors (ATRi) (Figure 1D). Since

ARID1A knockout cells already display elevated R-loops, is it possible that the impact of KLF5 is reduced? Is this the proposed model?

The limited effect of KLF5 loss in *ARID1A* KO cells upon ATRi is due to the synthetic lethality between KLF5 and ARID1A. In Figures 1e and 1f, we show that *ARID1A* KO cells are sensitive to losing KLF5 even without the administration of ATR inhibitor. This is in line with the CRISPR screen layout (Fig. 1a), where we propagated the cells following the sgRNA library transduction for nine days before starting the ATRi treatment, which is sufficient to show the effect of gene knockout on the fitness of WT versus *ARID1A* KO cells. In other words, *ARID1A*-deficient cells are already sensitive to losing KLF5 and it is difficult to generate more sensitivity when treated with ATRi.

Similarly, it is not clarified how the authors rationalize the notion that ATRi or Chk1 inhibitors (Chk1i) elevate R-loop formation/TRCs specifically in cells lacking KLF5 (Fig. 4A and D).

We show that KLF5 regulates the chromatin recruitment of BRD4. As BRD4 has a broad role in controlling RNAPII transcription elongation through interaction with P-TEFb and regulating RNAPII pause-release, loss of chromatin-bound BRD4 (mainly the long isoform) causes stalling of RNAPII. The increase in RNAPII stalling upon BRD4 loss in combination with ATR or CHK1 inhibitors, which are known to affect the replication machinery, leads to increased clashes between the transcription and replication machinery, accumulation of R-loops mainly at the TSS, and replication stress.

Curiously, based on Fig. 4A, transcription-replication collisions (TRCs) appear to diminish upon ATRi treatment in WT cells. The mechanism linking ATRi/Chk1i to TRC/R-loop formation, both in general and in cells lacking KLF5, should be discussed in more detail. For example, certain papers already report functional links between ATR and R-loops suppression (e.g., PMID: 37930853, PMID: 28314779).

This is an excellent point. Please see our detailed comments below (in specific points related to individual Figures; the second point related to Figure 4a).

The authors present an RNA-seq analysis of the transcriptome in cells lacking KLF5 versus WT. Beyond a somewhat limited pathway analysis, this data is not used further and appears disconnected from the rest of the study. In my opinion, it would be important to evaluate whether transcriptional dysregulation caused by KLF5 at specific genes is associated with abnormal formation of R-loops specifically at those genes.

We thank the reviewer for this comment and agree that it will be interesting to look for correlations between R-loops and transcriptional dysregulation by KLF5. We performed this analysis and found a statistically significant overlap between genes with dysregulated expression and genes with increased R-loops (see list of genes in the figure below and **new Supplementary Fig. 4g**). It is important to note the caveat that our differential expression analysis is between *KLF5* KO versus WT cells, and our differential R-loop analysis is between ATRi treated versus untreated *KLF5* KO cells. Additionally, as KLF5 promotes the recruitment of BRD4 in an acetylation-dependent manner mainly to enhancer regions, this presents another caveat for such analysis, as enhancers are not generally located within genes and instead regulate the

expression of distant genes. Thus, our results suggest the effect of KLF5 and BRD4 to be broad rather than specific to some genes. We have amended the manuscript and report this finding, **page 10 and 13**.

The figure above shows the genes that are differentially expressed, either upregulated or downregulated, in *KLF5* KO cells compared to WT cells, and show increased R-loop levels, based on DRIP-seq data.

Additionally, is the increase in R-loops caused by KLF5 knockout strictly observed at genes where BRD4 recruitment is dysregulated? Further analysis of the data in this regard is warranted: this could be achieved by a more thorough comparison between the DRIP-seq/RNA-seq data and the KLF5/BRD4 cut-and-tag experiments. As it stands now, the authors imply that dysregulated transcription in KLF5 knockout cells is responsible for elevated R-loop formation, but a direct link between these points is lacking. While inhibition of transcription is shown by the authors to reduce R-loops in KLF5 knockout cells, this would be expected regardless of whether KLF5 directly modulates R-loops at specific genes, or acts through other more indirect mechanisms.

We agree that this connection is an important point and is something we have previously attempted to address. However, it is challenging to identify genes directly regulated by BRD4 because BRD4 binds to enhancers, which are not generally located within genes and instead regulate the expression of distant genes. To determine these loci, we identified the genes located closest to the BRD4 binding sites. Although we can see that there is a significant increase in R-loop levels at these loci in the *KLF5* KO cells (see metagene profiles in new **Supplementary Fig. 6d**), this does not appear to be greater than the global impact we observe (Fig. 4e). This is potentially due to a more global impact of *KLF5* KO but could also be the result of the imperfect identification of BRD4-dependent genes.

Finally, it would also be of interest to check and comment on whether the expression of known R-loop regulators is modulated in KLF5 knockout cells versus WT at the transcriptional level (using RNA-seq data).

We agree with the reviewer about this point. Thus, we have checked the effect of KLF5 on known R-loop regulators at the transcriptional level using RNA-seq data in WT and *KLF5* KO cells. As shown in new **Supplementary Figure 7b**, KLF5 affects several R-loop regulators. We have also included additional data showing the effect of ARID1A loss on the transcriptional regulation of known R-loop regulators, which supported the mechanism of the synthetic lethality between ARID1A and KLF5. We further comment on this issue in the revised manuscript; **page 13-14**.

Specific points related for Figures:

-In Figure 3, it would be of interest to evaluate the incorporation of EdU (the levels/intensity of EdU incorporation and not only the fraction of cells that are EdU+) specifically in γ -H2AX high cells; one would expect that if lesions are occurring specifically in S phase, EdU incorporation might be reduced.

We thank the reviewer for this suggestion. In our revised manuscript, we have now included data showing the intensity of EdU incorporation in γ H2AX positive versus γ H2AX negative S-phase cells in new **Supplementary Figure 3b**. Indeed, as expected, γ H2AX high cells show less EdU incorporation due to the lesions caused by ATRi treatment, which affects productive DNA replication. Importantly, the percentage of γ H2AX high cells is significantly higher in *KLF5* KO cells, which leads to a more drastic effect on productive DNA replication and the ensuing hypersensitivity towards ATRi.

The authors should also comment on why cells would accumulate in G1, and not S or G2/M, if DNA lesions are accumulating during S phase in KLF5 knockout cells treated with ATRi. Additionally, the authors state, "KLF5 depleted cells accumulate more DNA damage and therefore withdraw at a higher rate from S-phase." It is not

clear what is meant by "withdrawing" from S phase and why damage in S would facilitate S phase progression/exit; a better discussion and explanation of these data and their significance is warranted.

We thank the reviewer for raising this point, and we now discuss this in a clearer way in the revised manuscript (**page 7**).

The most important observation is that *KLF5* KO cells have higher percentage of cells in S phase (EdU positive) when compared to WT cells in untreated conditions (**Fig. 3a-b**). This suggests that *KLF5* KO cells experience higher replication stress, which is sensed by intra-S-phase checkpoint, and thus progress slower through S-phase. However, treatment with ATRi (24 hours) would force both WT and *KLF5* KO U2-OS cells to progress through S and G2/M phases and appear to accumulate in G1 phase (Fig. 3a). Our interpretation agrees with previously published data showing that ATR regulates the intrinsic S-G2 checkpoint to ensure complete DNA replication before cells enter G2 phase (PMID 30139873). Moreover, a recent work has shown that ATR activity is required to support DNA replication in early S-phase (PMID 37336885), which is why cells treated with ATRi accumulate in G1.

Consistent with *KLF5* KO cells experiencing replication stress, we observed a drastic increase in γ H2AX positive signal in S-phase upon ATR inhibition (Fig. 3c-f). This ultimately results in increased genome instability (reflected in increased formation of micronuclei) and thus ATRi sensitivity.

To further substantiate these observations, we assessed the effects of *KLF5* loss on cell cycle profiles in untreated and ATRi-treated conditions in an additional cell line, RPE-1, as shown in **new Supplementary Figure 3a-b**. Consistent with what we observed in U2-OS cells, *KLF5* KO RPE-1 cells have higher percentage of cells in S phase (**new Supplementary Fig. 3a-b**). Moreover, we observed similar effects of *KLF5* loss in RPE-1 cells on the cell cycle distribution upon ATRi treatment, whereas RPE-1 WT cells showed a slight increase in the number of cells in G1 and no significant difference in the proportion of cells in S-phase upon ATRi treatment (**new Supplementary Fig. 3a**). Such a difference between parental U2-OS and RPE-1 cells might be because of different genetic backgrounds between these cell lines.

Finally, I note that the increase in DNA damage is limited in these cells (7-18% of cells are γ -H2AX high); as a control, it would be interesting to compare this with BRD4 siRNA or other conditions known to limit R-loops. Is this level of γ -H2AX accumulation expected for a bona fide R-loop regulator?

In the revised manuscript, we present additional data showing that using 2 μ M of ATRi for 24h, leads to a large increase in the % of cells with γ H2AX positive cells in both RPE-1 and U2-OS cells (**new Fig. 3e and Supplementary Fig. 3e-f**).

Nevertheless, based on the reviewer's suggestion, we tested the levels of γ H2AX accumulation upon depletion of either BRD4 or SETX, following ATRi treatment. Our results show that in the context of 1 μ M ATRi treatment, SETX depletion led to 5% of cells staining positive for γ H2AX, which is close to that of *KLF5* KO cells (7%), whereas BRD4 depletion caused slightly higher γ H2AX levels (10%), as shown in the figure below.

The figure above shows the percentages of γ H2AX positive cells in control, *KLF5* KO, SETX depletion or BRD4 depletion in untreated or ATRi treated settings.

-In figure 3, the authors report elevated γ -H2AX after 24h of ATRi treatment. I wonder if this could be due, at least in part, to cell death/apoptosis which can cause caspase-dependent elevation in γ -H2AX, which would change the interpretation of the data. Evaluating apoptosis (annexin V staining for example) and testing caspase-dependency of γ -H2AX signal would clarify this.

This is a very good point. In our revised manuscript, we now provide experimental data showing that 24h treatment with 1 μ M or 2 μ M ATRi did not cause a detectable increase in apoptosis in both the U2-OS and RPE-1 WT or *KLF5* KO backgrounds (**new Supplementary Fig. 3d,e**). Moreover, co-treating cells with ATRi and the caspase-inhibitor (Z-VAD) did not affect γ H2AX signals, suggesting that the observed phenotype is due to ATRi-induced DNA damage rather than caspase-dependent processes under our treatment conditions (**new Supplementary Fig. 3f-g**).

-In Figure 4A, a control in WT cells with ATRi and DRB is missing from these experiments.

We thank the reviewer for pointing this out. In our revised manuscript, we now include the ATRi/ CHK1i and DRB treatments in WT cells, **Figure 4a-b**.

It is also curious to see that ATRi reduces TRCs in WT cells. A statistical analysis of this should be presented and commented on if significant. This relates to a previous point regarding the need for the authors to at least briefly comment on why TRCs are elevated in *KLF5* knockout cells treated with ATRi or Chk1i but reduced in WT cells. This seems contradictory with published data indicating that ATR prevents R-loop generation via phosphorylation of key regulators, e.g., PMID: 28314779, PMID: 37930853.

We thank the reviewer for spotting out this curious phenotype in WT cells and for referring to those important papers. The reduction in TRCs in WT cells upon ATRi treatment is statistically significant, as presented in Figure 4a. The observed effect on TRCs in both WT and *KLF5* KO cells in untreated and ATRi-treated conditions could be explained by the following:

- To address the point of reduced TRCs in WT cells, we would like to draw the reviewer's attention to the fact that we performed PLA to detect TRCs in the

cells treated with ATRi, AZD6738, for 24h, while in the papers that the reviewer refers to, the authors used ATRi for either 30 min or 2h. Therefore, we speculate that during the early timepoints, there might be an increase in TRCs upon ATRi in WT cells; however, proficient R-loop regulation in WT cells might result in a reduction at the end point, 24h post ATRi treatment.

- To address the issue of increased TRCs in KLF5 KO cells, we observed an increase in R-loops (measured by DRIP-seq and RNaseH1 D210N) in both WT and KLF5 KO cells upon ATRi, yet the increase in KLF5 KO cells is far more pronounced (**Fig. 4c-e and Supplementary Fig. 4a-c**). As R-loops are considered a source of TRCs, this correlates with the elevation of TRCs in KLF5 KO cells.

Additionally, the RNA-seq data show that KLF5 regulates the expression of a subset of R-loop regulators, among them DHX9 whose expression appears to be reduced upon KLF5 depletion (**new Supplementary Fig. 7b**). This supports the observed increase in R-loops and TRCs in KLF5 KO cells.

Finally, we show that the chromatin-binding of BRD4 is disrupted upon KLF5 depletion, thus leading to increased R-loops. However, how BRD4 and other R-loop regulators and factors involved in the resolution of TRCs, such as DHX9 and DDX19, cooperate to resolve TRCs and prevent the formation of genotoxic R-loops is unknown and would be an interesting area for future studies.

In Figure 4, the authors show elevated R-loops in cells lacking KLF5. It is worth considering whether the increase in TRCs and R-loops might be linked to increased cell death. Is this a specific consequence of KLF5 loss, or is it due to lethality? One possible way to evaluate this would be to use an equitoxic dose of ATRi in WT cells to assess the impact on R-loop formation.

We provide several lines of evidence that the increase in TRCs and R-loops is a consequence of depleting KLF5 rather than cell-death related processes. Firstly, in the original manuscript, we presented data showing that removing R-loops by overexpressing WT RNaseH1 rescues *KLF5* KO cell lethality upon ATRi treatment. This result suggests that R-loops are the cause rather than a consequence of *KLF5* KO cells being hypersensitive to ATRi.

Moreover, in the revised manuscript, we present new data showing that treating cells with ATRi for 24h does not cause an increase in apoptosis (**new Supplementary Fig. 3c-d**). These results further confirm that under these ATRi-treatment settings to detect TRCs and R-loops, the observed phenotypes are due to loss of KLF5 rather than cell death processes.

To further support this conclusion, we assessed the impact of using equitoxic doses of ATRi in WT (2 μ M) and KLF5 KO (1 μ M) cells on R-loop formation. As shown in the figure below, treating WT cells with 2 μ M ATRi led to a significant increase in R-loops when compared to untreated WT cells. However, this increase was not significantly different than the increase in *KLF5* KO cells treated with half the dose, 1 μ M ATRi. These results therefore suggest that the combination of KLF5 loss and ATRi at low doses exacerbates the effect on R-loop formation, which is equivalent to the effect of double the ATRi dose in WT cells. These results are important, as they show that

using a lower ATRi dose affects *KLF5* KO cells more than WT cells, which could have clinical relevance.

The figure above shows the mean intensities of RNH1(D210N)-GFP as a measurement for R-loop formation in control and *KLF5* siRNA-depleted cells upon ATRi treatment at the indicated concentrations.

-In figure 4: the increase in TRCs in *KLF5* KO treated with ATRi vs DMSO is not supported by a statistical test (Fig. 4A).

In our revised manuscript, we now include a statistical test between *KLF5* KO ATRi vs DMSO, **Figure 4a**.

-Figure 5A and B appear redundant; one of the panel could be in supplemental.

We agree with the reviewer. In our revised manuscript, we have moved Figure 5a of our original submission to the supplementary information (**new Supplementary Figure 5a**).

-The input (total lysate) should be presented in Fig. 6A and S5A.

In our revised manuscript, we now include whole-cell extract blots, showing the protein levels of BRD4 in WT and *KLF5* KO in the different cell lines: see new **Figure 6a (bottom blot)** and **Supplementary Figure 6a (right blot)**.

-In figure 6E: a statistical test between *KLF5* knockout (KO) with and without siBRD4 would be appropriate to determine if the difference is significant. The current grouping of samples for statistical purposes is not very informative regarding the point being made.

We fully agree with the reviewer, and in our revised paper, we have modified the statistical test in Figure 6e. There is no statistically significant difference between *KLF5* KO with and without BRD4 siRNA depletion, as evidenced by the p-values provided in Figure 6e.

Minor points:

-Page 2: "This also increases conflicts between the replisome and the transcription machinery, associated with formation of DNA-RNA hybrids (R-loops; 11, leading to genome instability and threatening cell survival through replication-fork collapse 12,13."; the parenthesis needs to be closed.

In our revised manuscript, we have modified this text.

-The last section's header is "KLF5 loss impairs BRD4 chromatin recruitment and is lethal for ARID1A-null cells", however, no explicit discussion of the synthetic lethality is made in this section.

We thank the reviewer for bringing this up. We have modified the revised manuscript accordingly.

Reviewer #4 (Remarks to the Author)

The manuscript by Awwad et al describes the discovery of KLF5 as a synthetic lethal interactor of ARID1A, and further shows that loss of KLF5 sensitizes cells to ATR inhibitors. Mechanistically, the authors show that treatment of KLF5-depleted cells with ATR inhibitors causes replication stress, transcription-replication collisions, and defective BRD4 localization. Overall, the manuscript will be a welcomed addition to the field, with important findings and convincing results.

We thank the reviewer for acknowledging the quality of our work and its utility.

Specific comments

1. While the manuscript is set up by a CRISPR screen identifying ARID1A as synthetic lethal with KLF5, after a brief validation of this genetic interaction, the authors focus on investigating KLF5 in the response to ATR inhibitors. There is somewhat of a disconnect here. I would also note that some aspects of the manuscript's novelty are impacted by previously-published findings that KLF5 promotes BRD4 localization, and that BRD4 is synthetic lethal with ARID1A. Thus, any additional mechanistic insights that can be provided would increase the manuscript's novelty. The authors could investigate if the synthetic lethality between ARID1A and KLF5 (as well as BRD4) correlates with replication-transcription conflicts, increased R-loops, increased gH2AX/RPA foci, and is suppressed by RNAseH1 overexpression.

We thank the reviewer for these comments and suggestions. Accordingly, we have performed more mechanistic investigations related to the synthetic lethality between ARID1A and KLF5. Firstly, we investigated the effect of KLF5 loss on ARID1A mutant cells using data available in DepMap studies. In the **new Supplementary Figure 1i** of our revised manuscript, targeting *KLF5* in *ARID1A*-deficient cancer cells has a negative CRISPR score. As a positive control, we examined the effect of targeting ARID1B, a known synthetic lethal partner of ARID1A. Moreover, we assessed the effect of other transcription factors: KLF4 as another member of the KLF family, and E2F7 as a more broadly used transcription factor. Targeting either KLF4 or E2F7 does not appear to influence *ARID1A* mutant cell lines. These analyses further strengthen

our observations and conclusions that KLF5 is a synthetic lethal target for ARID1A-deficient cells and implies the specificity of KLF5 in such settings.

Secondly, we have carried out RNA-seq in *ARID1A* KO cells and assessed the effect of both ARID1A and KLF5 on the transcriptional regulation of R-loop associated genes. As shown in **new Supplementary Figure 7b**, both ARID1A and KLF5 regulate a subset of R-loop regulators. Prompted by such analysis, we show that co-depleting ARID1A and KLF5 increases the γ H2AX signal (2.4 fold-increase), and that this signal is diminished upon RNaseH1 overexpression (2.1 fold-decrease) (**new Supplementary Fig. 7c**). Adding to this, we provide evidence that KLF5 is required for recruiting BRD4 to chromatin in ARID1A KO cells (**new Supplementary Fig. 7a**). Collectively, these results suggest that the synthetic lethality between ARID1A and KLF5 correlates with dysregulation of R-loop homeostasis and accumulation of DNA damage, resulting in cell death/slow cell proliferation.

2. The validation of the screen results is performed only in U2OS cells. Additional results need to be presented in the support of the statement in the last sentence of the Abstract (KLF5 is a target for ARID1A-deficient cancers). The authors should be able to easily confirm the synthetic lethality in other cell lines (they already have KLF5-knockout RPE1 cells). They can also use the KLF5 inhibitor (ML264) to investigate if this causes increased cell death in ARID1A-deficient cells.

We thank the reviewer for making these suggestions. As shown in our revised manuscript, we have been able to demonstrate the synthetic lethality using cell proliferation assays in another cell line, RPE-1, and showed that *ARID1A*-deficient cells are sensitive to siRNA-mediated knockdown of KLF5 compared to WT cells (**new Supplementary Fig. 1d**). Also, we assessed the effect of a KLF5 inhibitor on the proliferation of U2-OS and MCF-7 *ARID1A* WT and KO cells. As shown in **new Supplementary Figure 1e-f**, KLF5 inhibition significantly reduced the growth of *ARID1A* mutant versus *ARID1A* WT cells. Furthermore, we took advantage of the triple negative breast cancer cell line, CAL-51, which does not express ARID1A, and complemented these cells with an empty vector or a vector expressing ARID1A-WT (**new Supplementary Fig. 1g**). Complementation with WT ARID1A rescued the slow proliferation observed upon KLF5 inhibition in CAL-51 cells, as presented in **new Supplementary Figure 1h**.

It should be noted that observed reduction in the growth of ARID1A mutant cells treated with KLF5 inhibitor is not large, yet is significant, and might be related to the mode of action, specificity and other factors affecting the efficiency of the KLF5 inhibitor used (ML264).

3. In my view, a significant weakness of the paper is that the majority of the mechanistic work (Fig 3-6) was performed in a single cell line, using a single KLF5 knockout clone (U2OS KO5). In addition, a subset of the presented experiments are inconsistently performed with siRNA (gH2AX after ATRi in 3C is done with KO while CHK1i in 3D is done with siRNA). Some of the experiments in these figures need to be shown with multiple KO clones (or both KO and siRNA) and in additional cell lines, to enhance the rigor of the study.

In response to these suggestions, we have validated the CRISPR screen results in an additional cell line. Thus, we generated HAP-1 *KLF5* KO clones and showed that such clones are hypersensitive to ATRi (**new Fig. 2d and Supplementary Fig. 2e**).

Moreover, in our revised manuscript, we now include additional data in additional cell lines to enhance the rigor of our study:

1. We show that *KLF5* loss has similar effects on cell cycle profiles in untreated and ATRi-treated conditions in RPE-1 cells, **new Supplementary Figure 3a-b**.
2. We confirmed the increase in S-phase γ H2AX signal upon both ATRi and CHK1i in U2-OS *KLF5* KO cells, **new Figure 3c-f**.
3. We show that ATRi treatment results in an increase in the percentage of γ H2AX cells in RPE-1 *KLF5* KO cells compared to control cells (**new Supplementary Fig. 3f**).
4. We also show that ATRi treatment results in an increase in the percentage of cells with replication catastrophe upon ATRi treatment (using different ATRi concentrations), in both U2-OS and RPE-1 cells (**new Supplementary Fig. S5b-c**).

Collectively, these data confirm the observed phenotypes in another cell line, with multiple *KLF5* KO clones, suggesting that the effect of *KLF5* is not cell line specific, but rather has a wider effect and sheds further insight into the consistency of our results.

4. Confirming the results in additional cell lines is also important in light of the fact the observed impact of *KLF5*-KO, albeit statistically significant, appears minimal. For example, through multiple figures, the % cells with γ H2AX staining and the % cells with RC go up from 1% to 5%. Is this increase sufficient to explain the loss of viability? It seems somewhat unlikely. Does increasing the ATRi concentration or the treatment time result in a more drastic increase/difference?

We agree with the reviewer, and in a revised manuscript have now included new results confirming the observed phenotypes in RPE-1 *KLF5* KO cells, as presented in **new Supplementary Figure 3f and Supplementary Figure 5b-c**.

Related to % cells with γ H2AX staining and % cells with RC, we presented data in the original manuscript related to both treatment time (ATRi; 2h versus 24h) and drug concentration (CHK1i; 100nM versus 200nM), which confirms that increasing the time or concentration results in a more drastic difference between WT and *KLF5* KO cells.

To further substantiate these results, we now include new results where we increased ATRi concentration (2 μ M for 24h), which results in a more drastic increase in % γ H2AX positive cells and % cells with RC in both U2-OS and RPE-1 cell lines (see **new Fig. 3d-f, Supplementary Fig. 3f-g, and Supplementary Fig. 5b-c**).

In addition, we draw the reviewer's attention to work (PMID: 32741974) where the authors show in Figure 5D that 24h treatment with ATRi (AZD6738) in *ATM* KO cells resulted in around 8% of cells with RC. Importantly, as *ATM*-deficient cells are hypersensitive to ATRi, this % of cells with RC seems to be sufficient to explain the loss of viability.

Furthermore, it should be noted that ATR inhibitors inhibit the kinase activity of ATR rather than directly inducing DNA damage. Therefore, we think that treating cells with ATRi would not cause a massive increase in γ H2AX signal, but that this marker of

DNA damage accumulates as the cells progress through S-phase of the cell cycle, where ATR activity is predominantly needed.

5. There are no statistical analyses shown in Fig 3A.

We now add statistical analysis in Figure 3a of the revised manuscript. Also, we have included additional data regarding the effect of KLF5 depletion on cell cycle distribution upon ATRi treatment in RPE-1 cells (**new Supplementary Fig. 3a**).

Reviewer #5 (Remarks to the Author)

We thank reviewer 5 for taking part in the review process for our paper.

Response to the reviewers

We very much appreciate the time and efforts that the reviewers and the editor have taken in evaluating our revised manuscript, and we are delighted that our paper is accepted, in principle, for publication in Nature Communications.

Our responses are in the order that they were raised in the reviewers' reports, with reviewers' comments in black and **our responses in green**.

Reviewer #1 (Remarks to the Author):

The authors have addressed my concerns. However, I would advise adding two references PMID: 33986538, 38754421 to the discussion in lines 421 "supported by ARID1A's function in modulating R-loops 54".

We thank the reviewer for their positive response on our revised manuscript, and we now cite those important papers in our manuscript.

Reviewer #2 (Remarks to the Author):

Reviewer #3 (Remarks to the Author):

The authors have answered my concerns satisfactorily, and in my opinion the current version is suitable for publication.

We appreciate that the reviewer finds our work suitable for publication.

Reviewer #4 (Remarks to the Author):

The revised manuscript addresses my comments and, in my opinion, the other reviewers' comments as well. The authors added a large number of additional experiments that validate their initial findings and reveal new mechanistic insights underlying the observed synthetic lethality.

We thank the reviewer for acknowledging the amount and quality of the additional experiments we have added to validate our initial findings, and we appreciate that the reviewer finds our work suitable for publication.